# Nascent liver proteome reveals enzymes and transcription regulators under physiological and alcohol exposure conditions

Jiayu Gu [1,6], Lihui Lao [2,6], Linzhen Hu [2], Jia Zang [2], Chao Liu [2,3], Ruixi Wan [2], Ling Tang [2], Ying Yuan [1] ✉, Yulin Chen [1,2,3] ✉ & Shixian Lin [1,2,3,4,5] ✉

The liver proteome undergoes dynamic changes while performing hundreds of essential biological functions. Dysregulation of the liver proteome under alcoholic conditions leads to alcohol-associated liver disease (ALD), a major health challenge worldwide. There is an urgent need for quantitative and liver-specific proteome information in living animals to understand the pathophysiological dynamics of this largest solid organ. Here, we develop a comprehensive approach that specifically identifies the nascent proteome and preferentially enriches membrane proteins in living mouse hepatocytes and is broadly applicable to studies of the liver under various physiological and pathological conditions. In the ethanol-induced liver injury mouse model, the nascent proteome successfully identifies and validates a number of transcription regulators, enzymes, and protective chaperones involved in the molecular regulation of hepatic steatosis, in addition to almost all known regulatory proteins and pathways related to alcohol metabolism. We discover that Phb1/2 is an important transcription coregulator in the process of ethanol metabolism, and one identified fatty acid metabolism enzyme Acsl1/5, whose inhibition protects cells and mice from lipid accumulation, a key symptom of hepatic steatosis.

The liver is the metabolic center of the human body and performs essential biological functions to conserve and regulate a wide range of metabolites[1,2]. Hepatocytes, accounting for approximately 80% of the total liver mass, harbor a variety of biological pathways that maintain the physiological homeostasis of the liver[3]. Hepatocytes dynamically express and secrete a large number of liver-specific proteins in response to various metabolic states under physiological conditions. In addition, hepatic impairment due to hepatocyte dysfunction (e.g., fatty liver) or reduced hepatocyte numbers (e.g., liver cirrhosis) leads to significant alterations in the liver proteome[2,4–7]. Alcohol-associated

liver disease (ALD) is the most prevalent type of chronic liver disease worldwide[4]. More than half of all liver-related deaths are attributed to ALD. With continued exposure to alcohol, ALD develops from a range of histological lesions, including alcohol-related hepatic steatosis, steatohepatitis with hepatic inflammation, and cirrhosis[2,6]. Therefore, there is a pressing need to understand the pathophysiological dynamics of the liver proteome in order to dissect the liver functions and intervene in liver diseases. In recent years, mass spectrometry (MS)-based proteomics has revealed the cytological profiles of the human liver proteome[8,9], proteomic biomarkers for alcohol-related

[1]Department of Medical Oncology, Laboratory of Cancer Prevention and Intervention Ministry of Education, The Second Affiliated Hospital of Zhejiang University School of Medicine, Life Sciences Institute, Zhejiang University, Hangzhou, China. [2]Zhejiang Provincial Key Laboratory for Cancer Molecular Cell Biology, Life Sciences Institute, Zhejiang University, Hangzhou, China. [3]Shaoxing Institute, Zhejiang University, Shaoxing, China. [4]Institute of Fundamental and Transdisciplinary Research, Zhejiang University, Hangzhou, China. [5]State Key Laboratory of Transvascular Implantation Devices, The Second Affiliated Hospital, Zhejiang University School of Medicine, Hangzhou, China. [6]These authors contributed equally: Jiayu Gu, Lihui Lao. ✉e-mail: yuanying1999@zju.edu.cn; chenyulin@zju.edu.cn; sxlin@zju.edu.cn

liver disease[10,11], early-stage liver cancer[12,13], and drug exposure[14]. Despite these advances in proteome elucidation using acutely isolated liver cells, methods are still lacking to gather liver-specific proteome data in living animals so as to investigate the molecular mechanisms of liver regulation and function under various pathophysiological conditions.

In contrast to homeostasis proteome analysis, nascent proteome analysis is of particular interest because of its ability to provide a sensitive and timely identification of the dynamically changing key regulator proteins. A variety of methods are available to selectively analyze newly synthesized proteins for a sensitive and timely understanding of the dynamic changes of the proteome in cultured cells[15–21]. However, with the exception of stochastic orthogonal recoding of translation (SORT)[22] and bioorthogonal noncanonical amino acid tagging (BONCAT) strategies[23,24], most of these methods cannot be used to discover the nascent proteome in specific tissues of living animals. The BONCAT strategy, pioneered by the Tirrell group in 2006, introduced the ability to study nascent proteomes with temporal resolution[16]. The most commonly used noncanonical amino acids in BONCAT are the methionine analogs, azidohomoalanine and azidonorleucine (ANL)[25]; notably, ANL-BONCAT enables cell-specific in vivo tissue analysis[23,26]. Building on BONCAT, researchers have extensively explored protein synthesis dynamics across tissues and cell types, revealing critical roles in metabolic regulation[27], protein stability[28], and neuronal activity–dependent responses[29] with tremendous successes. SORT was developed based on genetic manipulation of aminoacyl-tRNA synthetases in specific cell types of living animals followed by labeling of the nascent proteome with noncanonical amino acids. Labeling the nascent liver proteome in living animals is challenging due to the metabolic complexity of this largest metabolic organ, the presence of a large amount of pre-existing protein contamination, and the lack of specific and efficient labeling methods. This task is even more challenging in diseased animal models (e.g., ethanol-induced liver injury mouse model), where the timing of labeling is difficult to select and the accessibility of the diseased tissue is poor. As such, we sought to develop a comprehensive in vivo labeling strategy to decipher the nascent liver proteome in living mice with high efficiency and temporal resolution.

For this purpose, we develop the Stochastic Orthogonal Recoding of Translation induced by AAV-delivered Cre (SORT-AC) system. We show that SORT-AC preferentially enriches liver membrane proteins and is suitable to analyze acute and subtle changes in the nascent liver proteome under various pathophysiological conditions without dissection. After applying SORT-AC to the ethanol-induced liver injury mouse model, we identify and validate Phb1/2, a transcription coregulator for alcohol metabolism, Acsl1/5, a potential target for alleviating alcohol-induced lipid accumulation, and almost all known regulatory proteins and pathways associated with alcohol metabolism.

## Results

### Labeling the nascent proteome, including membrane proteins

We surveyed current proteomic labeling techniques applicable to complex organisms[19,22,27,28,30] in search of methods adaptable to the nascent liver proteome. We were particularly interested in labeling the nascent membrane proteome because the liver membrane proteome is likely to respond directly to changes in the external environment and no such approach has been reported in the literature[31,32]. Considering the complexity of the metabolism and the protein contamination already present in the liver, we assessed the method used for the nascent liver proteome based on the following criteria: (1) high spatial and temporal resolution for liver-specific labeling; (2) high sensitivity for efficient labeling of the low-abundance of the newly synthesized liver proteome; (3) good stability and bioorthogonality of the labeling probes in the most metabolically complex organ; (4) good accessibility and enrichment of the labeling probes in the liver; (5) capable of the

nascent membrane proteome labeling. Among the available methods, both BONCAT and SORT are uniquely capable of identifying the nascent proteome from spatially and genetically defined regions of living animals. We selected SORT for further optimization in our study, primarily because its substrate, AlkK, is readily synthesized at a multigram scale in our laboratory. An advantage for conducting large-animal experiments where substantial amounts of noncanonical amino acid probe are required. However, the SORT method has not been applied to label nascent liver proteome, probably because of its low labeling efficiency in hepatocytes. To improve the efficiency, we first sought to identify an efficient aminoacyl-tRNA synthetase, for the incorporation of AlkK (Fig. 1a), an inexpensive and stable noncanonical amino acid probe in liver cells[22]. The AlkK probe contains an alkyne handle that can be labeled by an azide-containing probe though click chemistry for subsequent microscopic imaging and biochemical enrichment (Fig. 1a). Since the Pyrrolysyl-tRNA Synthetase/Pyrrolysyl-tRNA (PylRS/PylT) pair is widely used for the incorporation of a variety of lysine derivatives and is broadly orthogonal in *E. coli*, mammalian cells, and mice, we rationalized a collection of PylRS variants for efficient AlkK recognition (Fig. S1a). Using GFP with an amber codon (TAG) at the 190 site as a reporter, we found that PylRS variant 1, carrying the Y384F mutation in PylRS[33], had the highest AlkK incorporation efficiency, with an amber codon read-through efficiency of approximately 50% relative to wild-type GFP (Fig. 1b, c and Supplementary Fig. 1b), at least fivefold higher than the variant (PylRS variant 4) previously used for residue-specific AlkK incorporation (Fig. 1b, c and Supplementary Fig. 1b)[22]. LC-MS analysis verified the fidelity of AlkK incorporation (Fig. S1c). We further engineered PylRS to optimize the incorporation efficiency by constructing a chimera[34], adding a nuclear export signal (NES) peptide sequence[35], and generating a four-unit tandem PylT$_{CUA}$ cascade (4 × PylT$_{CUA}$) (Fig. 1d). The engineered AlkKRS/4 × PylT$_{CUA}$ pair significantly increased the incorporation efficiency compared to the original pair, as determined by flow cytometry (Fig. 1e and Supplementary Fig. 1d, e).

To assess the residue-specific incorporation of AlkK throughout the proteome, we used a strategy called SORT to randomly display AlkK in the proteome of mammalian cells. SORT allows the proteome to be labeled at a large set of sense codons simultaneously by engineering the anticodon of the corresponding tRNAs. We reckoned that since membrane proteome turnover of hepatocytes readily respond to external metabolic changes[31,32], the dynamics of the nascent membrane proteome would be highly informative for dissecting liver function. However, labeling methods for the nascent membrane proteome have not been previously reported. To optimize AlkK incorporation into the entire nascent proteome with a preferential enrichment of membrane proteins, we chose the targeting residue for noncanonical amino acid incorporation based on the following rationale: these residues exhibit (1) high distribution within the total proteins, (2) notable enrichment within transmembrane proteins, (3) high distribution in protein surface instead of interface. Therefore, we first compared the distribution of 20 amino acids in total proteins and membrane proteins (Fig. 1f, g)[36–38]. Meanwhile, we also evaluated the proportional distribution of amino acids on protein surfaces and interfaces (Fig. 1f, g)[39], with the thinking that surface-displayed AlkK would be more suitable for labeling and enrichment. In the end, we chose the following representative residues as targets, PylT$_{UUU}$ (K-AAA), PylT$_{UGC}$ (A-GCA), PylT$_{GCU}$ (S-AGC), PylT$_{CAU}$ (M-ATG), and their combination PylT$_{UUU/UGC/GCU/CAU}$ (hereafter named PylT$_{KASM}$) (Fig. 1f–h). Among them, K, A, and S are abundant in the proteome and enriched at the protein surface (Fig. 1f and Supplementary Fig. 1f), while S and A are relatively abundant in the membrane proteome (Fig. 1g). M is chosen for total proteome labeling because it is the initial amino acid of all proteins (Fig. 1f and Supplementary Fig. 1f). Labeling with an azide-PEG4-biotin probe in the cell lysate and immune-blotting with streptavidin-HRP (SA-HRP) showed that the PylT$_{KASM}$ labeling

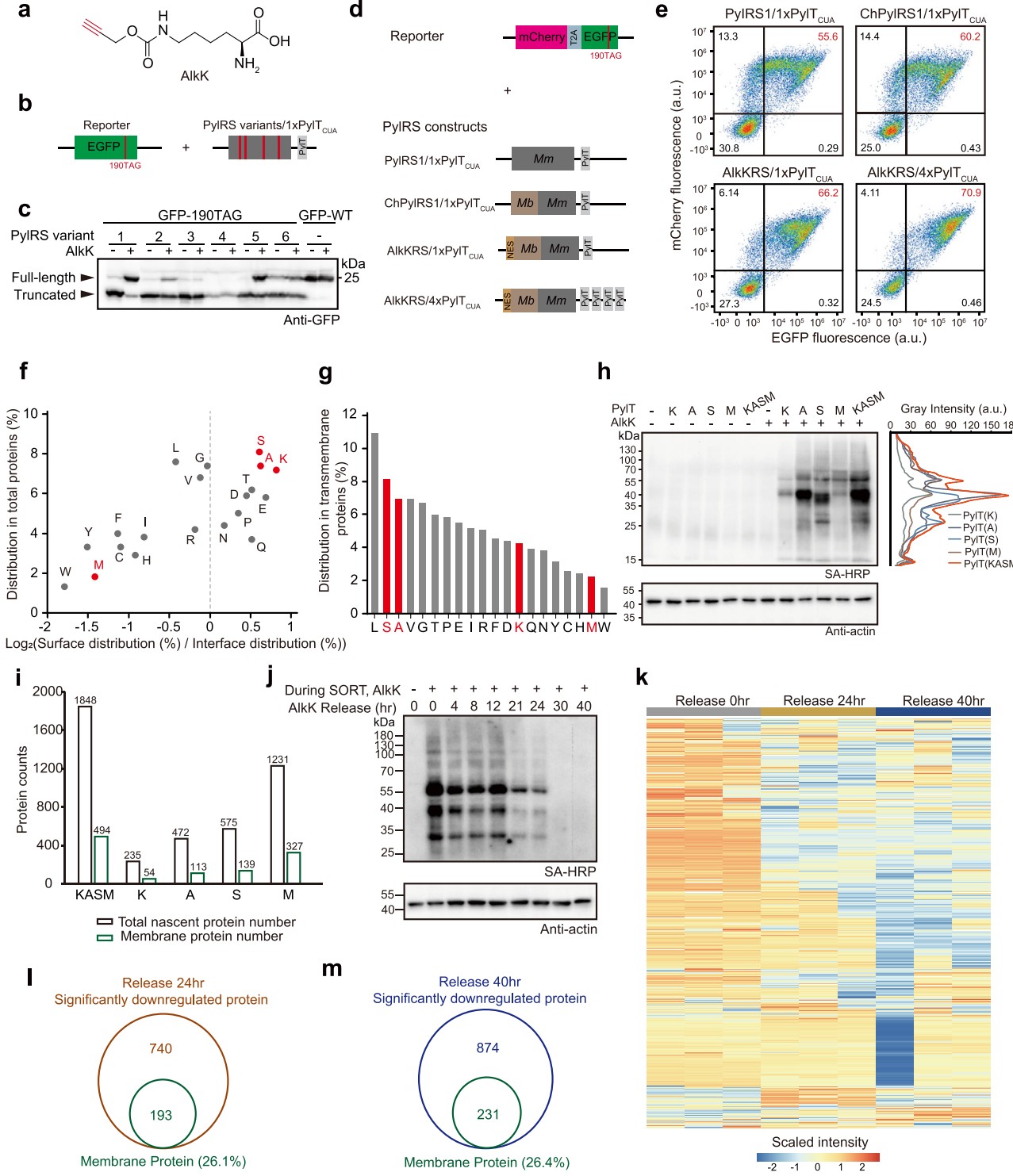

group was AlkK-dependent and more efficient than the group using the combinations of several identical PylTs (Fig. 1h). Subsequent LC-MS/MS analysis of nascent proteomes, following streptavidin enrichment, further confirmed that this combination of residues increased both total protein and membrane protein identification (Fig. 1i and Supplementary Fig. 1g).

To further evaluate this strategy, we assessed nascent proteome dynamics and overall performance (Supplementary Data 1). AlkK release experiments in cells revealed a rapid decrease in nascent protein signals, nearly diminishing within 30 h of AlkK withdrawal (Fig. 1j). Subsequent LC-MS/MS analysis of enriched nascent proteins

(Supplementary Fig. 1h) showed a significant reduction in protein abundance at 24 and 40 h post-AlkK removal (Fig. 1k and Supplementary Fig. 1i). Specifically, 740 and 874 proteins were significantly downregulated at these time points, respectively, with membrane proteins constituting 26.1% and 26.4% of these changes (Fig. 1l, m).

## Labeling the nascent proteome in mouse liver

To label and identify proteins expressed in the liver at specific time points, we extended the above system to live mice (Fig. 2a). A homogeneous SORT$_{KASM}$ mouse model was established by CRISPR-Cas9 knock-in of the loxP-flanking stop sequence at the H11 locus near the

**Fig. 1 | The strategy for labeling nascent proteins in prokaryotes and eukaryotes. a** Chemical structure of AlkK with reactive handle colored in red. **b** Cartoon of vectors used in the GFP reporter assay for amber suppression efficiency. GFP-190TAG reporter was co-transformed with PylRS variant into *E. coli* and full length of GFP is visualized in the presence or absence of AlkK with western blot analysis. **c** Western blot analysis of the amber suppression efficiency in *E. coli* with PylRS variants. Arrows indicate the full-length and truncated GFP. Experiment was independently repeated three times. **d** Schematic of vectors in the FACS reporter assay, in which the sequence from *Mm*PylRS is gray, the sequence from *Mb*PylRS is brown, the nuclear export signal sequence is yellow. The PylRS vectors was co-transfected with the mCherry-T2A-EGFP reporter into cells for the subsequent analysis. **e** FACS analysis of amber suppression efficiency in HEK293T cells transfected with the orthogonal systems with 1 mM AlkK. **f** Analysis of 20 natural amino acids distribution on total vertebrate proteins, and the ratio of amino acids distribution on surface and interface on proteins. The residues selected for AlkK incorporation are

colored in red. **g** Distribution of 20 natural amino acids in human transmembrane proteins. The residues selected for AlkK incorporation are colored in red. **h** SORT labeling of AlkK across the whole proteome of mammalian cells with the indicated anti-codons. Right, the quantitative analysis of labeling signal intensities. Experiment was independently repeated three times. **i** Analysis of nascent proteomes labeled in cells transfected with PylT-K, -A, -S, or -M individually or in combination. **j** Nascent proteomes harvested at time points from 0 to 40 h after AlkK removal from HEK293T cells transfected with SORT system and cultured with AlkK for 48 h. Experiment was independently repeated three times. **k** Heatmap analysis of changes in the nascent proteome at 0, 24, and 40 h after AlkK release following SORT labeling. **l, m** Quantification of significantly downregulated total proteins and membrane proteins at different time points. The significance downregulated parameter of S0 = 0.1 and a false discovery rate <0.075. Source data are provided as a Source Data file.

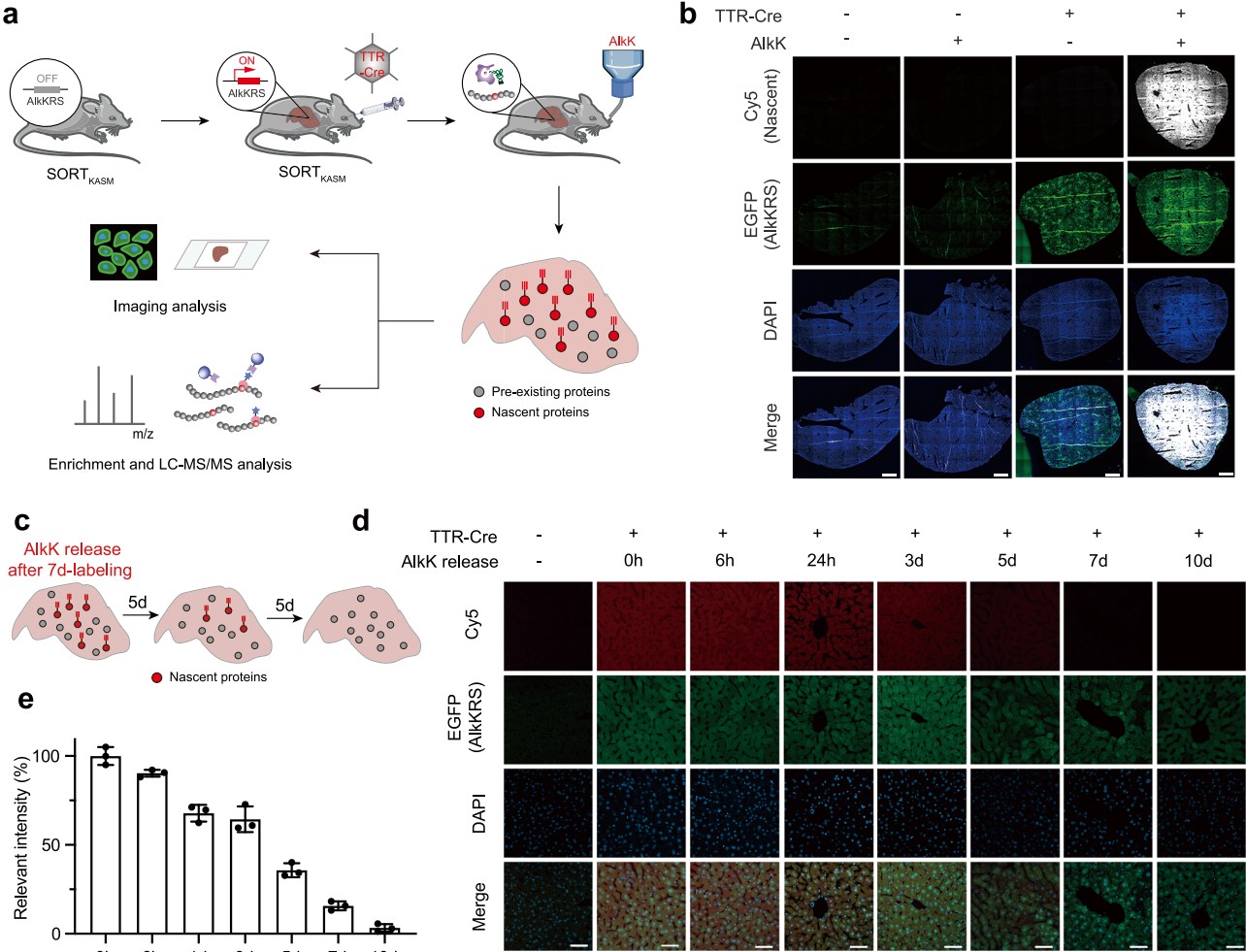

**Fig. 2 | The strategy for labeling nascent proteins in living mice. a** The strategy for labeling and identifying nascent proteome in SORT_KASM mouse liver by SORT-AC. Each mouse received an injection of 100 μL AAV-TTR-Cre (1 × 10^12 vg/mL). **b** The whole slide fluorescence scanning of liver sections. EGFP signal represented the AlkKRS expression level in liver tissue. Cy5 signal represented the labeled nascent proteome. Scale bar, 1 mm. Representative images of *n* = 3 experiments. **c** Schematic illustration of nascent proteome turnover in mice. SORT_KASM mice were administered AAV and allowed free access to drinking water supplemented with AlkK for 7 days. Following AlkK withdrawal, liver samples were collected at

designated time points from day 0 to day 7 for analysis. **d** Immunofluorescence analysis of mouse liver sections at different time points after AlkK withdrawal. Cy5 signals represent newly synthesized proteomes and show a gradual decline over time following AlkK removal. Scale bar, 100 μm. **e** Quantification of Cy5 fluorescence intensity shown in panel (**d**). For each group, liver sections from two mice were analyzed. Three random fields per sample were imaged, and the Cy5 signal intensity was measured and normalized to the 0 h time point. Data are the mean ± s.d.; *n* = 3. Source data are provided as a Source Data file.

chromosome 11 centromere, which expresses AlkKRS-P2A-EGFP under the CAG promoter and PylT_KASM under the U6 promoter, respectively (Supplementary Fig. 2a). When the packaged AAV containing the Cre recombinase gene under the hepatocyte-specific promoter TTR was

injected into SORT_KASM mice, Cre recombinase could excise the termination sequence (Fig. 2a and Supplementary Fig. 2a, b). As a result, AlkKRS was expressed in liver tissue (Fig. 2b and Supplementary Fig. 2c). Upon introducing AlkK into SORT_KASM mice via drinking water,

the SORT$_{KASM}$ system recognizes AlkK and randomly incorporates it into the newly synthesized liver proteins (Fig. 2a). Immunostaining of liver slices with an azide-Cy5 fluorescent probe showed that newly synthesized proteins in the liver were efficiently labeled by AlkK (Fig. 2b). Notably, we observed no significant changes in the body weight of mice expressing the SORT$_{KASM}$ system, and HE staining indicated that this system does not affect the normal histological functions of hepatocytes (Supplementary Fig. 2d). We named this strategy SORT-AC.

Furthermore, we extended our investigation of nascent proteome dynamics to the mouse liver using the SORT-AC strategy (Fig. 2c). We initiated labeling by injecting TTR-Cre packaged AAV into SORT$_{KASM}$ mice and providing AlkK in drinking water for 7 days. Subsequently, AlkK was withdrawn, and mice received normal drinking water for an additional 10 days (Fig. 2c). We observed a gradual decline in nascent hepatic proteome labeling signals after AlkK removal, with a noticeable reduction by day 1 and almost complete loss by day 10 (Fig. 2d, e). We determined the semi-quantified protein turnover half-life in mouse liver to be roughly 3–5 days. This result is consistent with the reported value of 3.28 days in the literature[40,41], indicating that SORT-AC is suitable for investigating the dynamic turnover of the proteome in vivo.

**Identification of liver-specific nascent proteome in living mice**
With SORT-AC mice, we set out to enrich for newly synthesized proteins in mouse liver under physiological conditions. The promoter of transthyretin (TTR) gene is sufficient to drive liver-specific expression of downstream genes, which has been validated in transgenic mice[42–44]. Based on this, we injected the SORT$_{KASM}$ mice with TTR-Cre packaged AAV and fed with or without AlkK in the drinking water for 7 consecutive days (Fig. 2a).

Although the immunostaining of liver slices showed high fluorescence intensity of the newly synthesized proteins, there was no significant increase of intensity in streptavidin-blot with azide-PEG4-biotin probe in lysate (Fig. 2b and Supplementary Fig. 3a), which might be due to the complicated environment of liver tissue. This discrepancy suggested interference from the complex biochemical environment of liver tissue. To test this hypothesis, we mixed SORT-labeled HEK293T cells with various mouse tissue extracts and performed the click reaction under identical conditions. A marked reduction in reaction signals was observed specifically in the liver and small intestine groups (Supplementary Fig. 3b, c). These findings indicate that the copper-catalyzed click reaction is readily inhibited or quenched in liver extracts, likely due to the high complexity of the hepatic biochemical composition (Fig. 3a). To improve the efficiency of click reaction in liver extracts, we optimized the conditions of the click reaction in three aspects, which are critical factors for efficient click reaction[45–47] (Fig. 3a): (1) the concentration of $Cu^{2+}$ and BTTAA ligands (Supplementary Fig. 3d), (2) the protein extraction method (Supplementary Fig. 3d), and (3) the reaction time (Supplementary Fig. 3e). The results showed that 2 h of reaction with 0.5 mM $CuSO_4$ and 1.0 mM BTTAA after methanol precipitation of liver extract led to the highest labeling efficiency of the nascent proteome (Supplementary Fig. 3f). Using this optimized reaction condition, we enriched nascent proteins using streptavidin beads (Fig. 3b). A large number of nascent proteins with a wide range of molecular weights were recovered with the addition of AlkK compared to the blank control (Fig. 3b).

By LC-MS/MS analysis, we identified a total of 1030 proteins in the enriched nascent proteome, including 61 unique proteins in the AlkK-supplemented group (SORT-AC Unique), 11 unique proteins in the control group (TTR-Cre injected but no AlkK supplemented), and 958 common proteins in both groups (Fig. 3c and Supplementary Data 2). Label-free quantification (LFQ) and $t$-test for 958 proteins in the common group showed that 233 proteins were highly enriched (SORT-AC Enrich) in the AlkK-supplemented group (Fig. 3d). Therefore, we combined SORT-AC Unique and SORT-AC Enrich to obtain 294

proteins newly synthesized by mouse liver in the physiological state, named as SORT-AC Print (Fig. 3e). Compared with the previous mouse tissue-specific proteomic dataset[48], more than 99% of the SORT-AC Print proteins were expressed in the liver, validating the high reliability of SORT-AC the analysis of the nascent liver proteome (Fig. 3f).

Furthermore, pathway analysis revealed that most of the SORT-AC Print proteins were related to amino acids and their derivatives metabolism, mono carboxylic acid metabolic processes, and cellular aldehyde metabolic processes, which are typical biochemical pathways for liver function (Fig. 3g and Supplementary Fig. 4a, b). Interaction network analysis revealed that proteins involved in proteasome assembly, protein catabolism and small molecule metabolism are actively synthesized in the liver under physiological conditions, including several liver marker proteins (Fig. 3h and Supplementary Fig. 4a). Interestingly, we found that SORT-AC enriched many membrane-associated proteins, accounting for ~50% of the total proteins (Fig. 3h and Supplementary Fig. 4a). As a result, 75 transmembrane proteins were identified among the 141 membrane-associated proteins in SORT-AC Print (Fig. 3i). These membrane proteins were distributed across all major membrane organelles and subcellular regions, in particular the plasma membrane, endoplasmic reticulum, mitochondrion, and Golgi apparatus (Fig. 3j). Analysis of 49 newly synthesized transmembrane proteins involving the endoplasmic reticulum, mitochondrion and Golgi apparatus (Fig. 3k) revealed that nearly 34% of these proteins are important components of lipid metabolism, which is one of the pivotal functions of liver tissue (Fig. 3l and Supplementary Fig. 4c). These results demonstrate that SORT-AC is a promising strategy for the identification of the liver-specific nascent proteome, especially the membrane-associated proteome, in living mice.

**Defining the liver-specific nascent proteome of ethanol-induced liver injury mouse model by SORT-AC**
Next, we applied SORT-AC to identify the nascent proteome under pathological states to explore the proteomic changes during the development of alcohol-related steatosis, which is directly caused by chronic excessive alcohol consumption[2,49]. To precisely capture the hepatocyte proteome response to acute alcohol stimuli, we implemented an acute alcohol gavage mouse model in conjunction with the SORT-AC labeling system (Fig. 4a). We gavage-fed mice with a high concentration of ethanol every two days and added a low concentration of ethanol to the drinking water after gavage to mimic both acute and chronic alcohol absorption patterns[50,51]. Preliminary experiments employing an intermittent gavage model successfully induced both body weight reduction and an inflammatory response in the murine liver (Supplementary Fig. 5a–d). Meanwhile, AlkK was dissolved in the drinking water to randomly incorporate into newly synthesized liver proteins. To ensure a consistent and quantifiable intake of AlkK, a preliminary study was conducted utilizing graduated drinking bottles to ascertain the approximate daily water consumption of the mice (Supplementary Fig. 5e). Mice were sacrificed after 27 days and liver samples were collected (Fig. 4a). Initial histological examination using Oil Red O staining revealed substantial areas of lipid accumulation within the liver tissue, suggesting the validity of our alcohol consumption modeling approach (Fig. 4b, c). We observed an elevation in serum AST levels in the mice, while ALT levels showed an insignificant increase, suggesting the onset of hepatocyte injury may be at an early stage (Supplementary Fig. 5g, h). Meanwhile, fluorescence results of liver tissue slices detected the expression of SORT-AC components and in situ labeling of the nascent proteome (Supplementary Fig. 5j). Liver protein extraction was then performed under the optimized conditions to enrich, verify and identify the nascent proteome (Supplementary Fig. 5i and Supplementary Data 3). Besides, the mice injected with AAV and supplemented with AlkK, but not treated with alcohol, were used for comparison (noEtOH). The mice without AAV,

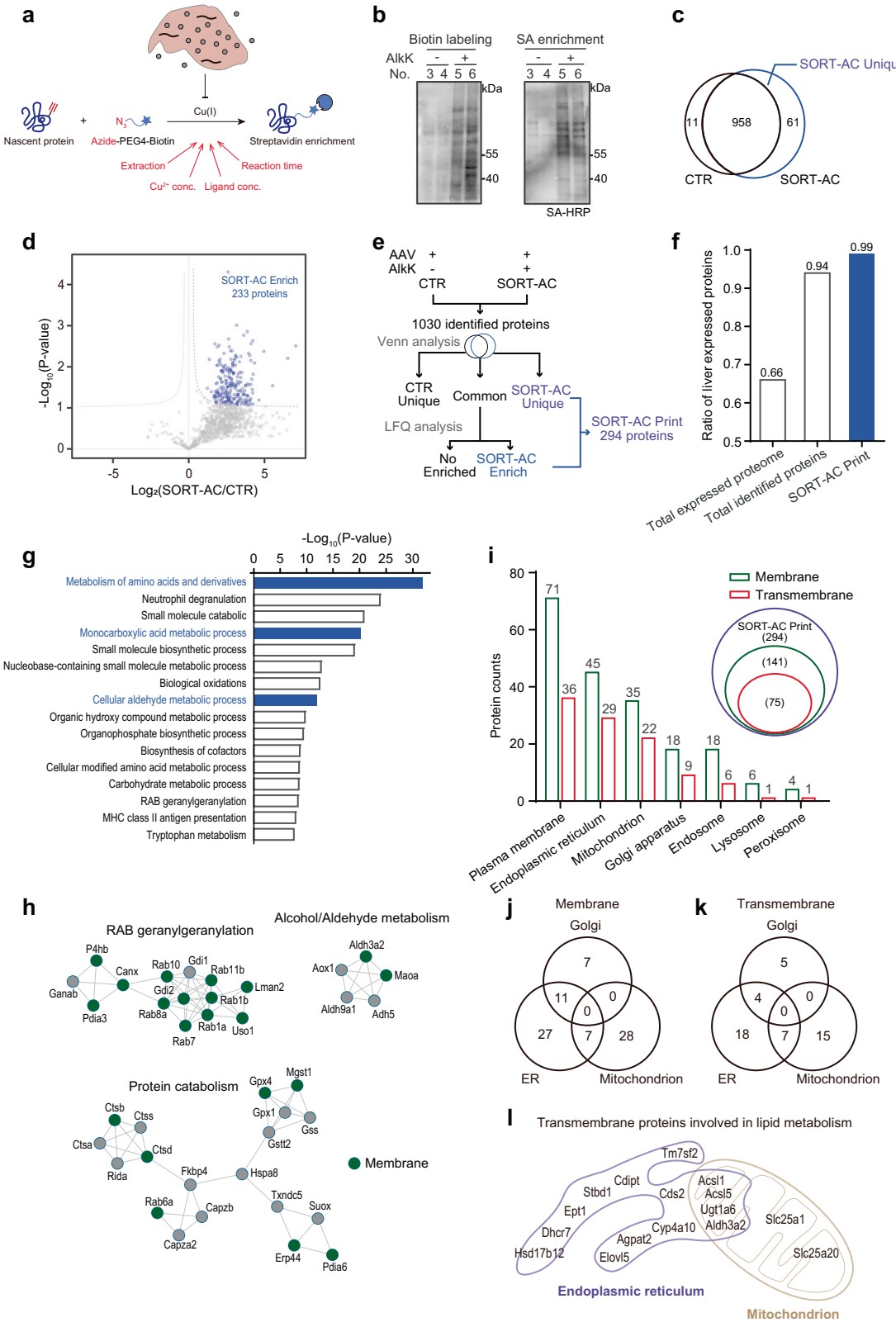

AlkK or alcohol were used as blank control for endogenous biotinylated contaminant proteins (CTR) (Fig. 4d).

Hierarchical clustering and principal component analysis (PCA) showed excellent reproducibility between different biological replicates, suggesting a good quality of sample preparation (Fig. 4e and Supplementary Fig. 6a). In addition, LFQ heatmap analysis identified a significant increase in the level and amount of newly synthesized

proteins in the ethanol-induced liver injury model with alcohol (EtOH group) compared to the group without alcohol (noEtOH group) (Supplementary Fig. 6b). Volcano plot significance analysis revealed that 187 proteins were enriched in the EtOH group (EtOH Enrich), indicating that these 187 proteins are highly upregulated and synthesized exclusively during alcohol-related steatosis genesis (Fig. 4f). As expected, the pathway analysis showed that these nascent proteomes

**Fig. 3 | Liver-specific nascent proteome identification and analysis in living mice. a** Nascent protein incorporated by AlkK containing an alkyne handle could be labeled by azide-PEG4-Biotin, and then be enriched by streptavidin. The reaction conditions, including extraction method, concentration of $Cu^{2+}$ and ligand, and reaction time, are optimized to improve the labeling efficiency in the liver extracts. **b** Labeling and enrichment of the nascent proteins of the whole proteome by western blot. SA-Blot, immunoblotting with streptavidin-HRP. Experiment was independently repeated three times. **c** Overlap of proteins identified between the control group and SORT-AC group. The proteins that were detected only in the SORT-AC groups were highlighted as SORT-AC Unique. **d** Volcano plot of the proteins identified in both the control group and SORT-AC group. The dashed lines were under the significance curve parameter of $S_0 = 0.1$ and a false discovery rate <0.05. The proteins significantly enriched were colored in blue and highlighted as SORT-AC Enrich. Statistical significance was assessed using a two-sided, empirical Bayes moderated $t$-test. **e** The strategy of identifying SORT-AC Print proteins. **f** Tissue specific analysis with previously reported dataset among total expressed proteome, total identified proteins, SORT-AC Print, respectively. **g** Enrichment analysis of SORT-AC Print proteins that major participate pathways with a one-sided Fisher's exact test. The blue indicated the pathways highly related to liver functions. The resulting $p$-values were adjusted for multiple hypothesis testing using the Benjamini–Hochberg procedure. **h** The interaction network analysis of SORT-AC Print proteins. The membrane proteins were colored in green. **i** The ratio of membrane and transmembrane proteins in SORT-AC Print (top) and the annotation of the subcellular localization of SORT-AC Print membrane proteins (bottom). **j** Overlap of membrane proteins in SORT-AC Print in mitochondrion, endoplasmic reticulum, and Golgi apparatus. **k** Overlap of transmembrane proteins in SORT-AC Print in mitochondrion, endoplasmic reticulum, and Golgi apparatus. **l** Annotation of transmembrane proteins involved in lipid metabolism in SORT-AC Print, which were located in mitochondrion and endoplasmic reticulum. Source data are provided as a Source Data file.

were highly correlated with monocarboxylic acid metabolic process as well as biological oxidations and cellular aldehyde metabolic process, which are known to be associated with ethanol metabolism (Fig. 4g). In addition, proteins involved in endoplasmic reticulum protein processing, cellular stress responses, and fatty acyl-CoA biosynthesis were also significantly enriched, suggesting the potential importance of these pathways in the steatosis genesis (Fig. 4g). Consistent with this, interaction network analysis revealed substantial nascent synthesis of many essential proteins involved in ethanol metabolism, in addition to a large number of membrane proteins in the nascent proteome (Supplementary Fig. 6c).

Through the annotation of the UniProtKB/Swiss-Prot database[52], we identified 32 transmembrane proteins out of 101 membrane proteins in EtOH Enrich (Fig. 4h). In contrast to the nascent proteome in physiological liver tissues (Fig. 3i), the subcellular locations of membrane proteins in the nascent proteome in steatosis liver tissues were predominantly associated with the endoplasmic reticulum and mitochondria (Fig. 4i). These two organelles are considered to be the major intracellular organelles for alcohol metabolism. The upregulated newly synthesized proteins were then quantified in EtOH Enrich (Supplementary Fig. 6d). As expected, most of the proteins synthesized at high levels in EtOH Enrich were involved in transcription, xenobiotic metabolism, stress response and lipid metabolism (Fig. 4j). In summary, a number of proteins are upregulated in the liver during ALD genesis, and these proteins may be necessary for alcohol metabolism with unknown molecular mechanisms.

**Validating key regulators and enzymes involved in alcohol-induced lipid accumulation**

The cellular proteome changes in response to external alcohol stimuli, reflecting adaptive or responsive changes (Figs. 4j and 5a). To better understand this response, we performed an in-depth analysis of the significantly enriched nascent proteins in the EtOH group. This proteome includes several well-known enzymes involved in ethanol metabolism, including Adh, Aldh, Cyp, and Ugt family proteins (Supplementary Fig. 6c). Although a number of newly synthesized proteins have been identified under alcohol exposure conditions, the transcription regulation of these proteins remains poorly understood[53]. Therefore, we searched in the identified nascent proteome for transcription regulators that might respond to the alcohol metabolism. Interestingly, Prohibitin 1 (Phb1) and Prohibitin 2 (Phb2), components of the mitochondrion membrane-associated transcription coregulator complex, were highly enriched in the nascent proteome (Figs. 4j and 5a). Phb1 has been reported to be a regulator of mitochondrial morphology, respiration, and the cell cycle[54–56]. It has been found to affect transcriptional activity either directly, through interactions with a range of transcription factors, or indirectly, through interactions with chromatin remodeling proteins, as a transcription

coregulator[57]. To verify these roles, we overexpressed either Phb1 or Phb2 in HepG2 cells and assessed downstream genes using quantitative PCR (qPCR) to quantify the expression levels of selected downstream alcohol-responsive genes. Consistent with the nascent proteome data (Fig. 4f), the mRNA levels of Adh1, Aldh3a2, Hspa5, Hsp90aa1, Acsl1, and Acsl5 were significantly upregulated after overexpression of Phb1 compared to control cells. Meanwhile, the mRNA levels of Adh1, Hspa5, and Hsp90aa1 were significantly upregulated after overexpression of Phb2 compared to control cells (Fig. 5b). These results suggest that Phb1 and Phb2 are two transcription coregulators that regulate the expression of alcohol metabolism genes.

We proceeded to validate protein targets enriched in the nascent liver proteome of ethanol-induced liver injury (Fig. 5a). Environmental stimuli are known to induce cellular stress responses and protein aggregation. Many Hsp family proteins were identified in EtOH Enrich (Fig. 4j), whose expression was also regulated by Phb1/2 (Fig. 5b). To validate the biological function of Hsp proteins in alcohol metabolism, we treated HepG2 cells with different concentrations of alcohol. The results showed that the level of protein ubiquitination correlated with the alcohol concentration, reflecting the alcohol-induced accumulation of misfolded proteins (Supplementary Fig. 7a)[58]. The concomitant use of Hsp70 (Apoptozole) and Hsp90 (17-AAG) inhibitors markedly exacerbated the accumulation of these protein ubiquitination and significantly reduced cell viability (Supplementary Fig. 7b–d)[59,60]. These findings suggested that the increased level of heat shock proteins in the nascent proteome serves as a protective response to counteract the detrimental effects of ethanol-induced protein misfolding (Fig. 5a).

Furthermore, in the nascent proteome, the acyl-CoA synthetase long chain family member 1/5 (Acsl1/5) is of particular interest because Acsl1/5 converts free long chain fatty acids into fatty acyl-CoA esters (Fig. 5a). Their functional role is well established in cellular uptake of fatty acids and remodeling, which influences various biological processes such as cell signaling, membrane synthesis, and energy homeostasis[61,62]. In addition, the fact that Acsl1/5 was also detected in the downstream of Phb1 (Figs. 4j and 5b) suggests an important role in alcohol metabolism via regulation of fatty acid metabolism. To explore this, we intervened in HepG2 cells using Triacsin C, a natural compound originally isolated from the fungus *Streptomyces aureofaciens*. It is a potent inhibitor of Acsl1/5, with a Ki value of approximately 15 nM[63,64]. Subsequent exposure of cell cultures to alcohol resulted in diminished lipid accumulation after pretreatment with Triacsin C, as confirmed by staining experiments and fluorescence-activated cell sorting analysis (Fig. 5c and Supplementary Fig. 7e, f)[65]. Additionally, we performed similar imaging and flow cytometry experiments in the AML12 cell line, confirming that Triacsin C can also inhibit ethanol-induced lipid accumulation in normal mouse cell lines (Supplementary Fig. 7g–i). Thus, Acsl1/5 is responsible for the lipid accumulation in

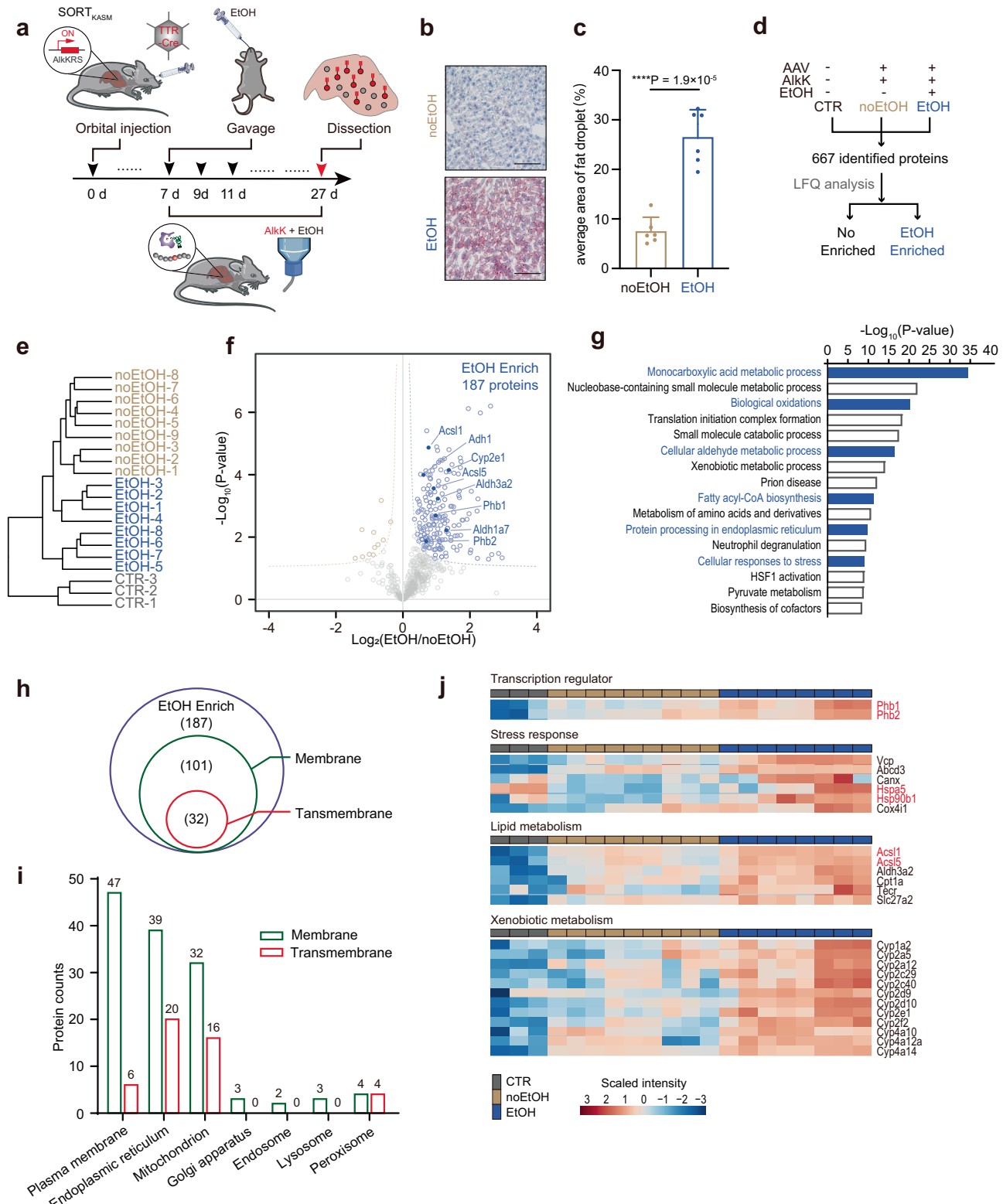

both HepG2 and AML12 cells during alcohol consumption. We further reasoned that application of Triacsin C treatment in a mouse model might alleviate the lipid accumulation and ethanol-associated cellular damage in hepatocytes. To test this in mice, we employed gavage administration of Triacsin C in an acute alcohol gavage mouse model (Fig. 5d). Subsequent serological analyses demonstrated a significant reduction in biomarkers (AST) indicative of hepatocellular damage (Fig. 5e, f), with no significant effect on serum triglyceride and ALT

levels (Supplementary Fig. 7j, k). Histological examination of liver sections utilizing Oil Red O staining revealed a diminished accumulation of lipids (Fig. 5g and Supplementary Fig. 7l). Importantly, administration of Triacsin C did not affect glucose metabolism[66] or induce a cytotoxic effect[67] in cultured cells, nor did it affect adrenal tissue or induce diarrhea in mice[68]. These results suggest that inhibition of Acsl1/5 by Triacsin C could alleviate ethanol-induced lipid accumulation in vitro and in vivo. Taken above results, we identified and

**Fig. 4 | Identification of the nascent liver proteome by SORT-AC in ethanol-induced liver injury mouse model. a** Schematic of the acute alcohol gavage mouse model establishing. In the workflow, AAV carried TTR-Cre (100 μL of $1 \times 10^{12}$ vg/ml per mouse) was orbital injected into C57BL/6J mice on day 0 and administered high-concentration ethanol gavage (4 g/kg) every two days. Meanwhile, a low concentration of ethanol (5%) and AlkK (30 g/L) was dissolved into the drinking water. The liver samples were collected from the mice after 27 days. **b** The Red Oil O staining of the mouse liver. The mouse without ethanol administered was used as control. The red regions indicated the fat droplets in which ethanol group has significant fat droplets accumulation. Scale bar, 100 μm. **c** Quantification analysis of (**b**). Data are the mean ± s.d.; $n = 6$ fluorescence images per condition. **d** The strategy of identifying EtOH-enriched proteins. **e** Hierarchical clustering of different indicated treatments. **f** Volcano plot of the proteins identified in both SORT-AC groups with or without acute alcohol gavage treatment. Statistical significance was

assessed using a two-sided, empirical Bayes moderated $t$-test. The dashed lines were under the significance curve parameter of $S_0 = 0.1$ and a false discovery rate <0.05. The proteins significantly enriched were colored in blue and highlighted as EtOH Enrich. **g** Enrichment analysis of EtOH Enrich proteins that major participate pathways, and the protein interaction network involved in monocarboxylic acid metabolic process. The blue bars indicated the representative pathways related to alcohol metabolism. GO term enrichment was performed using a one-sided Fisher's exact test. The resulting $p$-values were adjusted for multiple hypothesis testing using the Benjamini–Hochberg procedure. **h** The ratio of membrane and transmembrane proteins in EtOH Enrich. **i** The annotation of the subcellular localization of EtOH Enrich membrane proteins. **j** The heatmap analysis of EtOH Enriched proteins participated in transcription regulator, xenobiotic metabolism, stress responses, and lipid metabolism. Source data are provided as a Source Data file.

validated Phb1/2 as a transcription coregulator related to alcohol metabolism, while the protein ubiquitination is accumulated to elevated ethanol intake, and Acsl1/5 as a promising target for reducing alcohol-induced lipid accumulation.

## Discussion

In summary, we have developed a comprehensive approach, SORT-AC, that combines chemical biology tools with genetic models to analyze acute and subtle changes in the nascent liver proteome of living mice. SORT-AC consists of (1) AlkK for labeling newly synthesized proteins via click chemistry reaction; (2) AlkKRS/PylT$_{KASM}$ pair for efficient AlkK incorporation; (3) CRISPR-Cas9 knock-in mouse models for genomic integration of the AlkKRS/PylT$_{KASM}$ pair; (4) AAV-delivered Cre for rapid liver-specific expression of the AlkKRS/PylT$_{KASM}$ pair (Fig. 1i). We systematically optimized all these key components of the SORT-AC approach for liver-specific enrichment and identification of the nascent proteome, as well as for the investigation of the membrane proteome of the largest solid organ, in living animals. SORT-AC enabled efficient enrichment of the nascent membrane proteome by stochastic recoding the K, A, S, and M residues in the proteome. SORT-AC is generally applicable to quantitative analysis of the dynamic liver proteome under various physiological and pathological conditions with high sensitivity and confidence. Using SORT-AC in the ethanol-induced liver injury mouse model (Fig. 4a), proteins were identified with high-confidence under chronic and acute alcohol stimulation, including almost all known regulatory proteins and pathways associated with alcohol metabolism. The discovery of these proteins in a single experiment greatly expands the wealth of information on proteins and pathways related to alcohol metabolism, especially these newly identified membrane-associated proteins and pathways (Fig. 5a). More broadly, SORT-AC holds the hope to be a versatile approach for labeling and identifying the nascent proteome in potentially all animal tissues under various physiological and pathological conditions, provided that tissue-specific expression of Cre in AAV is achieved, like TTR-Cre for liver.

Finally, the identification of low-abundance transcription factors and enzymes using traditional MS-based methods is challenging. For example, despite decades of research, the transcription factors responsible for the metabolic response to alcohol are not well understood. In this study, we demonstrated that the ability of the SORT-AC approach to identify low-abundance regulators, including transcription regulators and key enzymes, during the development of alcohol-induced lipid accumulation. Previous transcriptomics-based studies[69–73] and traditional MS-based methods[8,11] did not indicate a significant upregulation of Phb1/2. SORT-AC revealed Phb1/2 as a functional transcription coregulator that enhanced the expression of enzymes and chaperone proteins involved in ethanol metabolism at the transcriptional level in the adaptive response of hepatocytes to alcohol. Furthermore, our study identified and validated Hsp family

proteins and many enzymes related to alcohol metabolism. Among them, Acsl1/5, which is also regulated by Phb1/2, is responsible for the lipid accumulation after alcohol consumption. An Acsl1/5 inhibitor had a notable inhibitory effect on alcohol-induced lipid accumulation in both cultured cells and mouse models. Therefore, our study has revealed that Acsl1/5 is a promising target for reducing ethanol-induced liver injury (Fig. 5e). Taken together, the discovery of key regulators in the ethanol-induced liver injury mouse model using SORT-AC paves the way for the discovery of low-abundance regulators and potential therapeutic targets in various disease models in the future.

## Methods
### General considerations

Unless otherwise noted, all commercial reagents were used without further purification. Primers and genes were synthesized by Tsingke Biotech. Anti-Flag affinity gel (20585ES08) was purchased from Yeasen, and Ni-IDA beads (SA003100) were purchased from SmartLifesciences. Primary antibodies: anti-GFP polyclonal antibody (2555, Cell Signaling Technology), anti-Actin monoclonal antibody (T40001, Abmart), anti-His polyclonal antibody (2365, Cell Signaling Technology), anti-Ubiquitin (10201-2-AP, Proteintech), and anti-Streptavidin-HRP antibody (3999, Cell Signaling Technology) were used at a dilution of 1:1000. Secondary antibodies: IgG mouse (SA00001-1, Proteintech) and IgG rabbit (M21002, Abmart) were used in a dilution of 1:5000. Transient transfections were performed using the Lip2000 reagent (BL623, Biosharp Life Sciences) and polyethyleneimine, linear, MW 25000 (23966, Polysciences).

The optical density at 600 nm wavelength ($OD_{600}$) and fluorescence intensity were acquired with a BioTek Synergy Neo2. Western blot was performed with a BioRad electrophoresis device, and chemiluminescence signals were captured by Azure Biosystems C400. FACS data were collected on a Beckman CytoFlex. LC-MS analysis was performed on a Xevo G2-XS QTOF MS system (Waters Corporation). LC-MS/MS analysis was performed on a Thermo Scientific Q Exactive HF-X Orbitrap and Orbitrap Exploris 480 mass spectrometer in conjunction with a Proxeon Easy-nLC II HPLC (Thermo Fisher Scientific) and Proxeon nanospray source.

GFP fluorescence for assessment of amber suppression efficiency was collected with Gen5 CHS 2.09 software. FACS of HEK293T cells was acquired with CytExpert (version 2.0.0.153, Beckman Coulter). Chemiluminescence of Western blot was captured by cSeries Capture Software (version 2.1.4.731, Azure Biosystems). LC-MS/MS raw files were acquired with Xcalibur (version 3.0.63, Thermo Fisher Scientific). The assessment of amber suppression efficiency was processed with Origin 2017pro software (version 9.4). Deconvolution of LC-MS spectra was performed using UNIFI software (version 1.9.4, Waters). FACS data were processed with FlowJo V10 (version 14.0.0.0, Flexera Software). LC-MS/MS raw files were analyzed with MaxQuant (version 2.2.0.0),

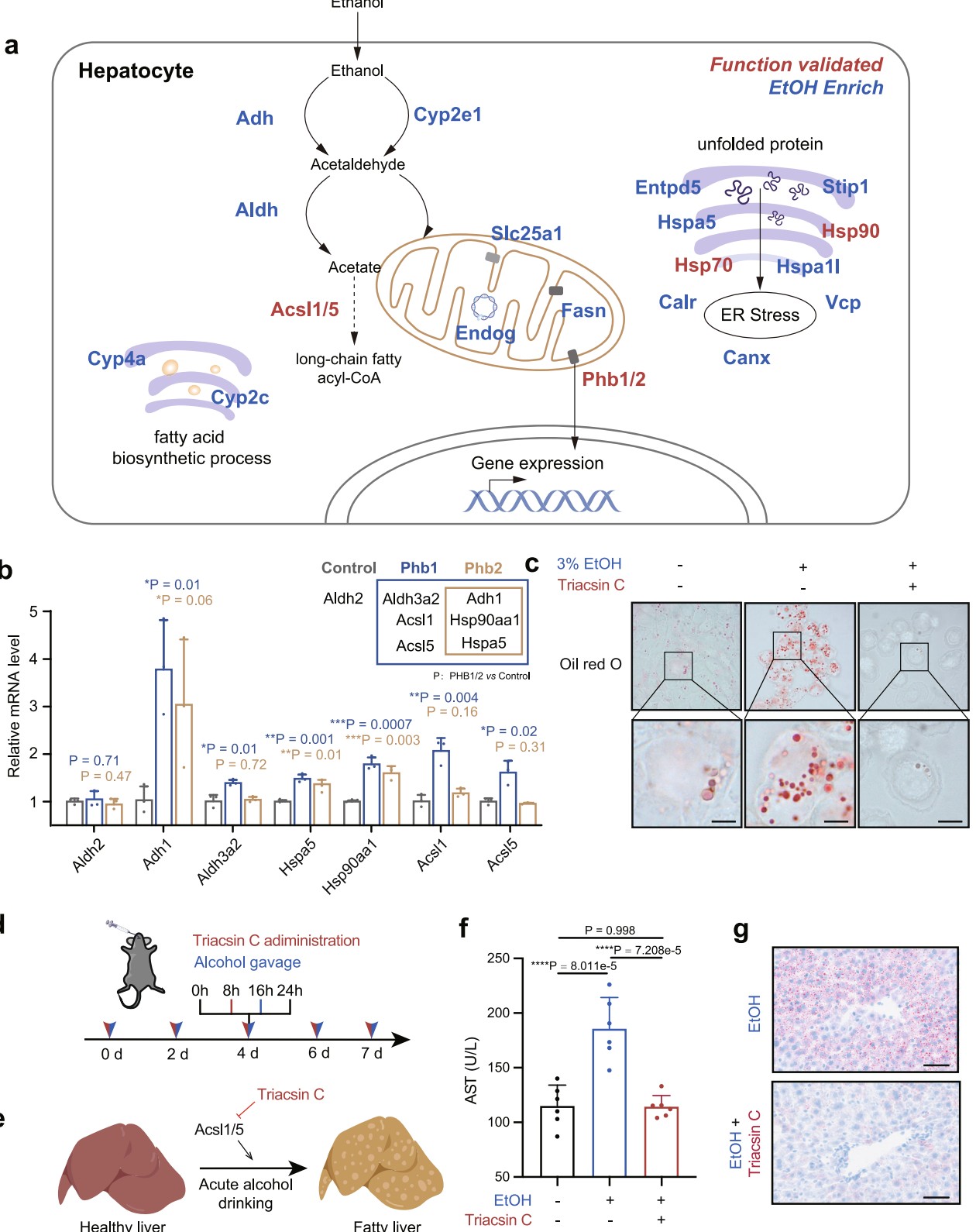

Perseus (version 2.0.7.0), Metascape web tool (version 3.5), Cytoscape (version 3.8.2), MetaboAnalyst web tool (version 6.0), R (version 4.2.0), and ggplot2 (version 3.4.2).

## Cell culture
HEK293T cells, HepG2 cells and AML12 cells from ATCC were maintained in an exponential growth as a monolayer in Dulbecco's modified Eagle medium (Thermo Fisher Scientific), high glucose, 10% fetal bovine serum (ExCell Bio) and 1% penicillin-streptomycin (Thermo Fisher Scientific), and cultured at 37 °C in 5% $CO_2$.

## Mice
All mice were reared in-house (temperature: 20–25 °C, humidity: 40–60%) in a 12 h light-dark cycle under specific pathogen-free

**Fig. 5 | Key regulators involved in ethanol-induced liver injury development. a** Overview the major pathways in hepatocytes response to alcohol exposure. EtOH Enriched proteins in SORT-AC were colored in blue and function validated proteins in this study were colored in red. **b** The mRNA expression levels of genes which regulated by Phb1 or Phb2 using qPCR (bottom) and schematic of the regulation relationship and inhibitor targets (top). Two-sided Student $T$-test was used to assess the significance of the differences between the corresponding Control group and the Phb1 or Phb2 group. Data are the mean ± s.d.; $n$ = 3 biologically independent repeats. **c** Triacsin C could effectively decrease lipid accumulation compared with the same ethanol treatment without Triacsin C using oil red O staining. Representative images of $n$ = 3 experiments. Scale bar, 10 μm. **d** Schematic of establishing the acute alcohol gavage mouse model with Triacsin C treatment. 4 g/ kg of ethanol was administered via gavage and after 8 h, 10 mg/kg of Triacsin C dissolved in 3% DMSO was given to the mouse by gavage every two days. After a week, the mice were fasted for 8 h, then anesthetized and blood was collected. **e** The schematic of Triacsin C function in protecting liver from lipid accumulation. **f** Analysis of the Triacsin C treatment group was significantly decreased the AST level. The normality of the data distribution was verified through the Shapiro–Wilk test. Statistical significance and $P$ values were determined using one-way ANOVA with Tukey's multiple comparison test. Data are the mean ± s.d.; $n$ = 6 mice. **g** The Red Oil O staining of the mouse liver. The mouse without Triacsin C administered was used as control. The red regions indicated the fat droplets in which ethanol group showed a significant accumulation. Representative images of $n$ = 6 mice each group. Scale bar, 100 μm. Source data are provided as a Source Data file.

conditions. All animals had free access to food and sterilized water. The care of the experimental animals was in accordance with the guidelines of, and approved by, the Institutional Animal Care and Use Committee of Zhejiang University.

In the investigation of nascent hepatic proteomics under physiological conditions, both female and male mice aged 8–10 weeks were included to more accurately represent the physiological state. For the study of alcohol-induced liver injury, only male mice aged 8–10 weeks were utilized to minimize the confounding effects of hormonal fluctuations associated with the estrous cycle.

## Plasmids construction

Unless otherwise stated, all plasmids were constructed by Gibson assembly. For the experiments in *E. coli* cells, the PylRS variants used for screening were cloned into pSupAR vectors. For the experiments in mammalian cells, chimeric PylRS variants and four tandem copies of PylT ($4 \times PylT_{CUA}$) were inserted into pCMV vectors. For labeling different amino acids experiments, $PylT_{UUU}$ (K-AAA), $PylT_{UGC}$ (A-GCA), $PylT_{GCU}$ (S-AGC) and $PylT_{CAU}$ (M-ATG) were mutated in pCMV vectors to $PylT_{CUA}$. Phb1 and Phb2 genes were amplified from HEK293T complementary DNA and inserted into pCMV vector carrying $3 \times$ Flag tag. Residue mutagenesis was introduced by Q5 Site-Directed Mutagenesis Kit (New England Biolabs).

## Chemical synthesis of AlkK

The AlkK was prepared according to the procedures[74]. For the synthesis of N6-((prop-2-yn-1-yloxy)carbonyl)-L-lysine (AlkK): To a solution of Boc-Lys ($2.0 \times g$, 8.1 mmol) in a mixture of 2 M sodium hydroxide (20 mL) and tetrahydrofuran (20 mL), the solution was stirred at 0 °C for 10 min. Propargyl chloroformate (0.63 mL, 6.5 mmol) was added in portions over a period of 5 min at 0 °C, and the resulting mixture was stirred at room temperature for 5 h. The reaction was washed with ice-cold diethyl ether (200 mL) and acidified with ice-cold 2 M HCl (200 mL). The aqueous phase was extracted with ice-cold ethyl acetate ($3 \times 50$ mL). The combined organic layers were dried over $Na_2SO_4$ and evaporated in vacuo to give a colorless oil. The carbamate was obtained in 75% yield without further purification, and subsequently deprotected with TFA in methylene chloride, delivering the amino acid AlkK as a white solid in 90% yield. 1H NMR (500 MHz, D2O) δ 4.68 (s, 2H), 3.54 (dd, J = 7.0, 5.5 Hz, 1H), 3.16 (t, J = 6.9 Hz, 2H), 1.86–1.67 (m, 2H), 1.61–1.52 (m, 2H), 1.46–1.34 (m, 2H).

## Assessment of amber suppression efficiency in *E. coli* and mammalian cells

The plasmid pBAD bearing the GFP-190TAG and the plasmid pSupAR carrying the corresponding synthetase and tRNA were co-transformed into chemically competent DH10B cells. The transformed cells were recovered in $2 \times YT$ medium for 1 h with shaking at 37 °C and plated on LB agar containing 30 μg/mL chloramphenicol and 100 μg/mL ampicillin for 12 h at 37 °C. A single colony was picked and grown in 2 mL of $2 \times YT$ containing the required antibiotics at 37 °C until the $OD_{600}$ reached ~0.8. Protein expression was induced by the addition of arabinose at a final concentration of 0.2% at 22 °C for 16 h with or without the addition of the AlkK. After induction, 1 mL of cell cultures were collected by centrifugation and then lysed with 150 μL BugBuster Protein Extraction Reagent (Millipore) for 20 min at room temperature. The supernatant of the lysate (100 μL) was transferred to a 96-well cell culture plate (Corning). The GFP signals of the supernatant were recorded by BioTek Synergy NEO2 with a background subtraction and normalized by the bacterial density ($OD_{600}$), which was also measured by BioTek Synergy NEO2 as well.

For FACS analysis in live cells, HEK293T cells were seeded in a 12-well plate and grown to 50–60% confluence for transfection. Cells were co-transfected with the pCMV vector and the pEGFP-mCherry-T2A-EGFP-190TAG at a ratio of 1:1 (μg:μg). Transfections were performed by lip2000 reagent (BioSharp) according to the manufacturer's protocol with or without the addition of the corresponding amino acids. At 48 h post-transfection, cells were trypsinized and neutralized by the complete medium before centrifugation. Cells were centrifuged at $200 \times g$ for 3 min, washed and resuspended in PBS for FACS analysis. The FACS instrument (Beckman CytoFlex) was set up according to the manufacturer's instructions. HEK293T cells were used to set appropriate forward scatter and side scatter gains. The fluorescent protein expressed cells were used to set FITC and PE gains and gate. At least 50,000 single cells were analyzed per condition. Finally, GFP fluorescence was acquired in the FITC channel, and mCherry fluorescence was acquired in the PE channel. FACS data were analyzed and processed with FlowJo V10.

## GFP expression, purification, and LC-MS analysis

GFP expression in *E. coli* was performed by the same procedure as mentioned above. After expression, the cells were collected by centrifuging at $3220 \times g$ for 15 min at 4 °C. The cell pellets were resuspended in ice-cold buffer A (50 mM Tris, 200 mM sodium chloride (NaCl), 50 mM imidazole, 1 mM β-mercaptoethanol, pH 8.0) and lysed by sonication. After centrifugation ($13,523 \times g$, 60 min), the lysate was loaded into an Ni-IDA column and then washed with 30 column volumes of buffer A. The proteins were eluted with 2 column volumes of buffer B (50 mM Tris, 200 mM NaCl, 250 mM imidazole, 1 mM β-mercaptoethanol, pH 8.0). Purified proteins were subjected to LC-MS analysis.

For LC-MS analysis for intact proteins, the purified proteins were analysed on a Xevo G2-XS QTOF MS system (Waters) equipped with an ESI source in conjunction with a Waters ACQUITY UPLC I-Class Plus. Separation and desalting were carried out on a Waters ACQUITY UPLC Protein BEH C4 Column (300 Å, $2.1 \times 50$ mm², 1.7 μm). Mobile phase A was 0.1% formic acid in water and mobile phase B was acetonitrile with 0.1% formic acid. A constant flow rate of 0.2 mL/min was used. Data were analysed using Waters UNIFI software. Mass spectral deconvolution was performed using UNIFI software (version 1.9.4, Waters). The molecular weight of the protein was predicted using the ExPASy Compute pI/ Mw tool (https://web.expasy.org/compute_pi/), and chromophore maturation in GFP was also considered in the calculation.

## Analysis of SORT-labeled proteome in HEK293T with click chemistry

SORT labeling in mammalian cells was performed as previously reported[75]. Briefly, HEK293T cells were seeded in a 6 cm dish and grown to 50–60% confluence. The cells were transfected with plasmids pCMV-AlkKRS-PylT$_{UUU}$(K), pCMV-AlkKRS-PylT$_{UGC}$(A), pCMV-AlkKRS-PylT$_{GCU}$(S), pCMV-AlkKRS-PylT$_{CAU}$(M) for the residue-specific incorporation of AlkK. The medium was replaced with fresh medium supplemented with 1 mM indicated AlkK at 6 h and 26 h post-transfection. After SORT labeling for 40 h, cells were washed, trypsinized, and pelleted. In studies of nascent proteome turnover, cells were co-cultured with AlkK for 40 h, followed by AlkK withdrawal. Cell samples were subsequently collected at 0, 4, 8, 12, 21, 30, and 40 h for further analysis.

All the samples were lysed with modified RIPA buffer (25 mM triethanolamine, 150 mM NaCl, 1% Triton ×-100, 0.5% sodium deoxycholate, 0.1% SDS, DNase I and RNase A/T1, pH 7.5) for 10 min at room temperature and centrifuged at 16,000 × $g$ for 15 min to collect the supernatant. Click chemistry reaction of samples was performed with 1 mM azide-PEG4-biotin (Alfa Aesar), 150 μM CuSO$_4$ (Sangon Biotech), 300 μM 2-(4-((bis((1-(tert-butyl)-1H-1,2,3-triazol-4-yl)methyl)amino)methyl)-1H-1,2,3-triazol1-yl)acetic acid (BTTAA, Sigma) and 5 mM ascorbic acid (Sigma) for 1 h at 30 °C and quenched with 1 mM bathocuproine sulfonate (Sigma) for 10 min. Then, 4 × SDS protein loading buffer and denatured at 100 °C for 5 min and analyzed by western blot.

For LC-MS/MS cell samples, after quenched with 1 mM bathocuproine sulfonate (Sigma), add an appropriate amount of streptavidin agarose beads (GenScript) washed three times with PBS. Then, protein samples were incubated with beads overnight at 4 °C on a rotator mixer. The following day, the samples were centrifuged for pellet separation. The pellet was washed sequentially with 500 mM NaCl/PBS twice, 0.2% SDS/PBS twice, PBS once, 2 M urea/PBS twice, and PBS three times. Finally, the beads were thoroughly dried for LC-MS/MS analysis.

For exploring the complexity of the mouse tissue, mice were first anesthetized, and tissues including heart, liver, lung, intestine, and kidney were collected. Appropriate amounts of each tissue were minced and lysed in a Modified RIPA buffer (25 mM triethanolamine, 150 mM NaCl, 1% Triton X-100, 0.5% sodium deoxycholate, 0.1% SDS, DNase I, and RNase A/T1, pH 7.5) supplemented with PMSF. Samples were centrifuged at 12,000 × $g$ for 15 min at 4 °C, and the supernatants were collected and stored for further use. The SORT labeling of cells was performed as described above. Briefly, HEK293T cells cultured in six-well plates at 50–60% confluency were transfected with the pCMV-AlkKRS-KASM plasmid. Six hours post-transfection, the medium was replaced with fresh culture medium containing 1 mM AlkK. After 40 h of labeling, cells were washed, trypsinized, and pelleted. The samples were lysed with a modified RIPA buffer for 10 min at room temperature, followed by centrifugation at 16,000 × $g$ for 15 min to collect the supernatant. Protein concentrations of tissue and cell lysates were quantified using the BCA Protein Assay Kit (Thermo Fisher). All samples were diluted to the same concentration (approximately 1 mg/mL) using modified RIPA buffer. Subsequently, 20 μL of cell lysate was mixed with equal volumes of lysates from different mouse tissues at matched protein concentrations. The mixtures were subjected to click chemistry reactions. Click chemistry was performed under the same conditions that had been successfully applied to labeled cells (1 mM azide-PEG4-biotin, 150 μM CuSO$_4$, 300 μM BTTAA, and 5 mM ascorbic acid at 30 °C for 1 h). The reactions were quenched by the addition of 1 mM bathocuproine sulfonate for 10 min. Samples were then mixed with 4× SDS protein loading buffer, denatured at 100 °C for 5 min, and analysed by dot blotting. The equal amount of the denatured samples was applied onto a nitrocellulose membrane, followed by SA-HRP-based detection. The labeled cells were used as a positive control to investigate the influence of the mouse tissue microenvironment on the performance of the click chemistry.

## Conditional knock-in mouse strain construction

The gene conditional knock-in mouse strains discussed in this article were constructed by Gempharmatech Co., Ltd. Utilizing CRISPR/Cas9 technology, SORT$_{AlkK-KASM}$ system was insertion at H11 site was achieved through homologous recombination. The guide RNA sequences and the mouse genotyping primer sequences are provided in Supplementary Data 4. Meanwhile, a loxP flanking stop sequence leading AlkKRS-P2A-EGFP is expressed from the CAG promoter, and PylT$_{KASM}$ is expressed from the U6 promoter. Parental heterozygous mice SORT$_{AlkK-KASM}$+/− are bred at a female-to-male ratio of 1:2 or 1:3. After the offspring are weaned, toe clipping for identification and tail snipping for genomic DNA extraction are performed. The experimental procedure follows the instructions provided in the Genomic DNA Isolation Mini Kit manual (Vazyme). Using designed primers, PCR amplification of the target genomic fragment is performed. Mice identified as homozygotes are cohoused for expansion breeding. The mice used in this article are littermate homozygotes.

## SORT-AC in live mice

Mice were anesthetized with 2% isoflurane and immobilized for orbital injection. Each mouse received an injection of 100 μL AAV2/8-TTR-Cre (1 × 10$^{12}$ vg/mL). Two weeks after the orbital injection, the drinking water for the experimental group mice was replaced with a beverage containing 30 mg/mL AlkK and 15% blackcurrant juice, while the control group mice received water without AlkK. Mice were allowed free access to water for one week. After 1 week of AlkK treatment, the mice were euthanized and perfused with ice-cold PBS and 4% paraformaldehyde fixative at a flow rate of 8 mL/min via cardiac perfusion. In experiments investigating the dynamic changes of the nascent proteome, mouse samples were collected at 0, 6, and 24 h, 3, 5, 7, 10, and 14 days after AlkK withdrawal. A portion of the liver was fixed in 4% paraformaldehyde for sectioning and imaging experiments, while another portion was flash-frozen in liquid nitrogen and stored at −80 °C for proteomic analysis.

Adequate amounts of tissue were combined with an equal volume of grinding beads and twice the volume of modified RIPA buffer modified RIPA buffer (25 mM triethanolamine, 150 mM NaCl, 1% Triton X-100, 0.5% sodium deoxycholate, 0.1% SDS, DNase I and RNase A/T1, pH 7.5), and homogenized using a cryo-grinder pre-chilled with liquid nitrogen for 3 min. The mixture was then sonicated for 3–6 min to promote protein solubilization, followed by centrifugation to collect the supernatant for protein quantification using a BCA Protein Assay Kit. To 1 mL of protein extract (-2 mg/mL), sequentially add 0.1 mM Biotin-PEG4-Azide, 0.5 mM CuSO$_4$, 1.0 mM BTTAA, and 5 mM sodium ascorbate, mix well, and allow to react at room temperature for 2 h, which was quenched with 1 mM BCS. The test conditions for this reaction were evaluated by immunoblotting (streptavidin-HRP) to display the biotin signal. Further, methanol was added to the reaction mixture to precipitate the protein.

An appropriate amount of streptavidin agarose beads (GenScript) was taken and washed three times with PBS. Then, protein sample were incubated with beads overnight at 4 °C on a rotator mixer. The following day, the samples were centrifuged for pellet separation. The pellet was washed sequentially with 500 mM NaCl/PBS twice, 0.2% SDS/PBS twice, PBS once, 2 M urea/PBS twice, and PBS once. Finally, the beads were thoroughly dried for LC-MS/MS analysis.

## LC-MS/MS of SORT-AC enriched nascent proteome

Streptavidin agarose beads enriched proteins were mixed with 4 × SDS loading buffer and denatured at 100 °C for 5 min. Samples were loaded into 10% SDS-polyacrylamide gel for electrophoresis. The region of the whole lane was cut out for proteomic identification. The samples were then subjected to reduction, alkylation, and digestion with trypsin (Promega) in gel. The resulting peptides were desalted by home-made tips with C-18 membrane and kept at −80 °C until analysis.

For the SORT-AC enriched nascent liver proteome at the physiology state experiment, peptides were analysed by LC-MS/MS on a Thermo Scientific Q Exactive HF-X Orbitrap mass spectrometer in conjunction with a Proxeon Easy-nLC II HPLC (Thermo Fisher Scientific) and Proxeon nanospray source. The desalted peptides were separated with a 75-micron × 100 mm Magic C18 200 Å 3 U reverse phase column and eluted using a 60 min gradient with a flow rate of 300 nL/min. An MS survey scan was obtained for the m/z range 400–1400 and resolution was set to 60,000 for full MS scan events with a maximum ion injection time (IT) of 45 ms. An automatic gain control (AGC) target value for full MS and MS/MS scan were set to $1 \times 10^5$ and $5 \times 10^3$, respectively. MS/MS spectra were obtained using a top 25 method, where the top 25 ions in the MS spectra were subjected to High Energy Collisional Dissociation (HCD). MS/MS scans were acquired with a resolution of 15,000 and maximum IT of 22 ms. An isolation mass window of 1.6 m/z was used for the precursor ion selection, and a normalized collision energy of 27 eV was used for fragmentation. A dynamic exclusion method was used for 30 s.

For the SORT enriched nascent cell proteome and the SORT-AC enriched nascent liver proteome in ethanol-induced liver injury model experiment, peptides were analysed by LC-MS/MS on a Thermo Scientific Orbitrap Exploris 480 mass spectrometer in conjunction with a Proxeon Easy-nLC II HPLC (Thermo Fisher Scientific) and Proxeon nanospray source. The desalted peptides were separated with a 75-micron × 100 mm Magic C18 200 Å 3 U reverse phase column and eluted using a 60 min gradient with a flow rate of 300 nL/min. An MS survey scan was obtained for the m/z range 400–1200 and resolution was set to 60,000 for full MS scan events with a maximum ion injection time (IT) of 50 ms. A normalised AGC target value for full MS and MS/MS scan were set to 300% and 100%, respectively. Precursor ions in the MS spectra were subjected to High Energy Collisional Dissociation (HCD). MS/MS scans were acquired with a resolution of 15,000 and maximum IT of 22 ms. An isolation mass window of 1.6 m/z was used for the precursor ion selection, and a normalized collision energy of 30% was used for fragmentation. A dynamic exclusion method was used for 30 s. All the LC-MS/MS raw files were acquired by Xcalibur (version 3.0.63, Thermo Fisher Scientific Inc.).

### Processing of LC-MS/MS based proteomic data

For the SORT enriched nascent cell proteome MS/MS raw files were processed with MaxQuant software[76] (version 2.2.0.0) with the integrated Andromeda search engine[77] against the Swiss-Prot human database (Release 2025-04-16) containing 20,421 reviewed entries. For the SORT-AC enriched nascent liver proteome MS/MS raw files were processed with MaxQuant software[76] (version 2.2.0.0) with the integrated Andromeda search engine[77] against Swiss-Prot mouse database (Release 2023-02-06) containing 17,145 reviewed entries. Trypsin/P specificity was required for the digestion mode, and the maximum allowed missed cleavages was set to 2. First search peptide tolerance was set to 20 ppm, and the main search peptide tolerance was 10 ppm. Peptides with a minimum of 7 amino acids and a maximum charge of 7 were considered. The required FDR was set to 0.01 at peptide and protein levels. Cysteine carbamidomethylation was set as the fixed modification and methionine oxidation was set as variable modification. Other parameters followed the default settings.

### Bioinformatic analysis of LC-MS/MS

For the SORT enriched nascent cell proteome, only the proteins quantified in at least 2 of 3 repeats of the sample condition were considered as identified in the respective sample condition, then for the following quantification analysis. The missing values were replaced from normal distribution by Perseus5 (version 2.0.7.0) with default parameters. The cell without AlkK was used as blank control for endogenous biotinylated contaminant proteins. SAM t-test were performed between release 0 h and release 24 h or release 40 h, with the

significance curve parameter of S0 = 0.1 and a false discovery rate <0.05. For the SORT-AC enriched nascent liver proteome at the physiology state experiment, only the proteins quantified in all the repeats of the sample condition were considered as identified in the respective sample condition. Proteins were separated into 3 groups: CTR Unique, Common, and SORT-AC Unique. Then quantification and SAM t-test were performed for the Common groups, with the significance curve parameter of $S_0 = 0.1$ and a false discovery rate <0.05. The significantly increased proteins in SORT-AC sample condition were named as SORT-AC Enrich. Then combination of SORT-AC Unique and SORT-AC Enrich was named as SORT-AC Print. For the SORT-AC enriched nascent liver proteome in ethanol-induced liver injury model experiment, only proteins quantified in at least 3 repeats in at least 1 sample condition were considered for the following quantification analysis. Then the missing values were replaced from normal distribution by Perseus[78] (version 2.0.7.0) with default parameters. The mice without AAV, AlkK, or alcohol were used as blank control for endogenous biotinylated contaminant proteins (CTR). SAM t-test were performed between noEtOH and EtOH conditions, with the significance curve parameter of $S_0 = 0.1$ and a false discovery rate <0.05, and significantly increased proteins in EtOH sample condition were named as EtOH Enrich.

LC-MS/MS bioinformatic analysis plots were displayed by R (version 4.2.0) with ggplot2 (version 3.4.2) packages. Pathway and interaction network analysis were performed by Metascape web tool[79] (version 3.5) and Cytoscape[80] (version 3.8.2). Hierarchical clustering, PCA, and heatmap were generated by MetaboAnalyst web tool[81] (version 6.0). Protein subcellular localization was annotated with UniProtKB/Swiss-Prot database[52].

### Immunofluorescence microscopy and confocal imaging of mouse liver sections

After fixation, the liver tissues were dehydrated in a 30% sucrose solution and submerged in OCT compound, followed by being frozen at −80 °C for 5–10 min. Sections of 6 μm thick were then cut using Cryostan NX-50 (Thermo Fisher Scientific). The cryosections were air-dried at room temperature, then were fixed, cleared, and sealed. The click reaction mixture (Cy5-Azide 10 μM, $CuSO_4$ 1 mM, BTTAA 2 mM, sodium ascorbate 5 mM) was applied to the tissue sections, allowed to react at room temperature for 2 h, followed by washing and mounting. Confocal images were captured via a Zeiss LSM880 and LSM900 super-resolution microscope and further analysed using ZEN software. To investigate the dynamics of the nascent proteome, images were acquired using identical parameters. Three fields of view were randomly selected for each sample, and Cy5 channel fluorescence intensity was quantified using ImageJ (version 1.52). The relative fluorescence intensity was calculated by normalizing each value to that of the 0 h sample, and the results were presented as bar graphs for analysis.

### Construction and characterization of a model for acute alcohol consumption

A preliminary experiment was performed to establish an ethanol-induced liver injury model using oral gavage of 4 g/kg ethanol every other day for a week, with an additional gavage administered one day prior to sample collection. Mouse body weight was recorded daily at a fixed time point (10:00 a.m.). On day 7, mice were anesthetized, and blood and liver tissue samples were collected. Serum was subjected to biochemical analysis for liver function markers, including ALT and AST. Liver tissues were fixed, paraffin-embedded, sectioned, and stained with HE for histopathological evaluation. In addition, total RNA was extracted from liver tissue using TRIzol™ reagent (Thermo Fisher Scientific), and reverse transcription was carried out using the HiScript® II Q RT SuperMix for qPCR (Vazyme) according to the manufacturer's instructions. Quantitative PCR primers were designed using PrimerBank (https://pga.mgh.harvard.edu/primerbank/), with

two independent primer pairs selected for each target gene to ensure specificity (Supplementary Data 4). qPCR reactions were prepared following the SYBR Green qPCR protocol (Vazyme) and performed on a Bio-Rad CFX96 real-time PCR detection system.

In SORT$_{KASM}$ mice, 4 g/kg ethanol gavage by every two days for 7 days with 5% ethanol added to the drinking water. Meanwhile, 30 mg/mL AlkK was administered in drinking water. After labeling for 16 days, the mice underwent cardiac perfusion and liver sectioning, and liver protein extraction as previously described. A portion of the liver tissue was cryosectioned and subsequently stained with Oil Red O and hematoxylin and eosin (HE). The stained sections were analysed using ImageJ (version 1.52). Another portion of the liver tissue was labeled for the nascent proteome, following the procedures mentioned above.

## qPCR of Phb1 and Phb2 transfected HepG2

HepG2 cells were seeded into a 6-well plate and grown until they reached 50–60% confluence for transfection. Plasmids pCMV-3 × Flag-PHB1 or pCMV-3 × Flag-PHB2 were co-transfected into the cells using Lip3000 transfection reagent (Biosharp), according to the manual. After 48 h, the cells were collected and RNA was extracted using TRI-zol™ reagent (Thermo Fisher Scientific). The extracted RNA was then reverse transcribed into DNA using the HiScript® II Q RT SuperMix for qPCR (Vazyme). Primers for qPCR were designed through PrimerBank (https://pga.mgh.harvard.edu/primerbank/), with three primer pairs selected for each gene. All the primers for qPCR are provided in Supplementary Data 4. Subsequently, the qPCR system was set up according to the SYBR Green qPCR Protocol (Vazyme) and run on a Bio-Red CFX96 real-time PCR detection system.

## Inhibitor of Hsp70 and Hsp90 treatment in HepG2

HepG2 cells were seeded in a 6-well plate and grown to 70–80% confluency. In order to investigate the impact of ethanol on protein folding, HepG2 cells were treated with a gradient concentration of ethanol ranging from 1% to 12%. Culture dishes were sealed with parafilm to prevent ethanol evaporation. After 1 h, cell samples were collected, and lysed in a NP-40 buffer (Biosharp) containing protease inhibitors (Roche) mixture on ice. After 20 min of centrifugation at $15,000 \times g$, the supernatant was collected, and 1% SDS solution was added to both the supernatant (non-aggregates) and the pellet (aggregates) for denature[82], for the subsequent western blot analysis. Additionally, after pre-treating the cells with 20 μM of Apoptozole (APExBio) and tanespimycin (MedChem Express) for 24 h, 5% and 10% ethanol were further added in cell medium for 8 h sealed with parafilm. Then the cell was collected for western blot analysis as above.

To test the effect of ethanol and inhibitors on cell viability, cells were treated with 10, 20, and 40 μM of Apoptozole and tanespimycin for 24 h, followed by treatment with 1, 5, and 7.5% ethanol for 8 h, sealed with parafilm. After staining with trypan blue, the number of viable cells was counted using a cell counter (Countstar).

## Triacsin C treatment in cells and mice

HepG2 cells and AML12 cells were seeded in a 6-well plate and reached 50–60% confluency. HepG2 cells were treated with 8 μM Triacsin C (Aladdin)for 3 h and 3% ethanol for the next 3 h sealed with parafilm. AML12 cells were treated with 10 μM Tracsin C for 24 h and 1% ethanol for next 24 h sealed with parafilm. For imaging, lipid level of these cells was evaluated by modified oil red O staining kit (Beyotime). After staining, the cells were observed and photographed under a microscope (Olympus).

To quantify the lipid droplets, Nile Red solution (MedChem Express) was added to the cells, and then incubated at room temperature for 30 min. The cells were washed with PBS and analysed with filter to detect fluorescence using a flow cytometer (Beckman). Fluorescent intensity was measured using FlowJo V10.

In mice, 4 g/kg of ethanol was administered via gavage and after 8 h, 10 mg/kg of Triacsin C dissolved in 3% DMSO was given to the mouse by gavage every two days. After a week, the mice were fasted for 8 h, then anesthetized and blood was collected. The whole blood was placed at 4 °C overnight, then centrifuged at $1000 \times g$ for 15 min to obtain serum. The serum was analysed for AST via Biochemistry Analyzers (Roche, COBASC 311). Additionally, the mouse livers were fixed and stained with Oil Red O staining kit, followed by microscopic analysis. The proportion of the area positive for staining was analysed using ImageJ (version 1.52).

## Statistics

Statistical analysis and data presentation were performed using GraphPad Prism 9 and R (4.2.0). Data and error bars in the figures represent the mean ± s.d. The normality of the data distribution was verified using the Shapiro–Wilk test. For comparisons between two groups, significance and $P$ values were calculated using a two-sided unpaired $t$-test for normally distributed data. For comparisons among three or more groups, statistical significance and $P$ values were determined using one-way ANOVA with Tukey's multiple comparison test for normally distributed data, or the Kruskal–Wallis test with Dunn's multiple comparison test for non-normally distributed data. The sample sizes ($n$) are indicated in the figure legends and the exact $P$ values are indicated in the figures (*$P < 0.1$; **$P < 0.01$; ***$P < 0.001$; ****$P < 0.0001$).

## Reporting summary

Further information on research design is available in the Nature Portfolio Reporting Summary linked to this article.

## Data availability

The mass spectrometry proteomics data have been deposited in the ProteomeXchange Consortium via the iProX partner repository[83] with the identifier PXD051725 (Nascent proteome reveals key regulators in the development of alcoholic liver disease). Previously published datasets[48] that were used for reanalysis of the tissue specific. Swiss-Prot human database (Release 2025-04-16) containing 20,421 reviewed entries and Swiss-Prot mouse database (Release 2023-02-06) containing 17,145 reviewed entries from UniProt was used for proteomics analysis. All the processed proteome data generated in this study are provided in the supplementary data files. Source data are provided with this paper.

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

## Acknowledgements

We are grateful to the core facility of the Life Sciences Institute and Prof. Xiangwei He for helpful discussions. We thank the National Key R&D Program of China (grant 2024YFC3407200), the National Natural Science Foundation of China (grant 22222705, 22437004, 92253302, 22361142828, and 82373415 for Y.Y.), the China Postdoctoral Science Foundation (2023M743033 for Y.C.), the Postdoctoral Fellowship Program of CPSF (grant GZC20232280 for Y.C.), the State Key Laboratory of Transvascular Implantation Devices (grant 012024012), the Fundamental Research Funds for the Central Universities (grant 226-2024-00046), the Zhejiang Provincial Natural Science Foundation of China (grant LZ24B020001), and the Feng Foundation of Biomedical Research for financial support.

## Author contributions

S.L. and Y.C. conceived the idea and supervised the study. J.G. and L.L. conducted all experiments and analysed the data together. L.H., J.Z., R.W., and L.T. assisted with experiments and provided valuable discussion. C.L. performed the chemical synthesis. Y.Y. co-supervised this study. S.L., J.G., L.L., and Y.C. wrote the manuscript. All authors commented on the final draft of the manuscript.

## Competing interests

The authors declare no competing interests.
