## [Transparent Peer Review file · Nature Communications]

Nascent liver proteome reveals enzymes and transcription regulators under physiological and alcohol exposure conditions

Corresponding Author: Dr Shixian Lin

Version 0:

Reviewer comments:

Reviewer #1

(Remarks to the Author)

The authors have developed a novel tool for the identification of tissue specific nascent proteome changes under different experimental conditions. This animal model is novel and very useful to the field. However, the authors have not done a full characterization of the dynamics of the nascent protein labeling in this study. Further, the incorporation of an alcohol model at the surface seems interesting but adds more questions since some important considerations were not made as detailed below. In summary I believe the animal model is novel and a great tool to investigate tissue specific nascent proteome but it needs more characterization. The way the alcohol model was performed I don't believe it is possible to interpret the results objectively because the no EtOH mice would have substantially more labeling as compared to the EtOH mice (e.g., the water only mice drink more than the 5% EtOH mice getting more AlkK).

Based on the concerns listed below I would recommend rejection with major revisions.

Major Concerns:

It is not clear if identified labeled membrane proteins were selected for analysis or if only membrane proteins could be labeled. The description of the protein labeling mechanism for the 4 amino acids doesn't support the notion that only membrane proteins will be labeled.

There is no methods section in the main body of the paper. This has to be included before a reviewer can assess or evaluate the manuscript.

The dose of the AAV needs to be included.

There needs to be rationale for the EtOH model used. Currently the model uses 3 EtOH binges followed by 20 days of with background consumption of 5% EtOH. It is unclear what the rationale for the 3 binges are if the mice are euthanized 16 days later. Further, the dynamics of the nascent proteome would be of interest. How many of the nascent proteins are sustained for the 20 day period post 1st binge?

There needs to be a more thorough investigation to the impact of AlkK dose needed for consistent labeling. Did all the mice drink the same amount of water getting the same dose of AlkK? Mice will drink 5% EtOH but will drink less than water alone which would alter the AlkK dose between experimental groups. That is one reason the Lieber DeCarli diet and the method of pair feeding is used so that the amount of diet is equal between the pair-fed and alcohol-fed mice.

An experiment should be performed to measure the rate of turnover of these nascent proteome in the liver. For example, provide the AlkK water for a week and euthanize a group of mice followed by replacing the water with no AlkK and euthanize mice daily until no nascent proteins are detectable. This is important characterization of the new animal model that is critical for future research.

HepG2s are not a relevant cell line to study ethanol exposure. Primary mouse hepatocytes or AML-12 cells are more applicable in vitro models.

There should be a more thorough phenotyping of the experimental mice other than liver enzymes and steatosis.

Minor concerns:

Title needs to be altered to remove "alcoholic". For example, Nascent liver proteome reveals novel enzymes and transcription regulators under physiological and conditions of alcohol exposure.

The raw proteome data is not found with the provided ID at the repository noted by the authors and the hyperlink requires a password. If the data is under embargo it needs to be lifted and verified.

Since EtOH is known to affect protein translation it may be important to note nascent proteins that were enriched in the No EtOH group vs EtOH group.

Individual data points are not shown for sample size n 10.

No statistic test are provided for figures (fig. 5f) and where they are presented they are not appropriate.

Reviewer #2

(Remarks to the Author)

This manuscript presents a proteomic method to identify newly synthesized proteins from the liver with a focus on membrane proteins. Their approach is the modification of a previous protocol termed SORT. Interesting novel biological findings were discovered when they applied their method to an EtOH induced injury mouse model. It was unclear, however, why this was a liver specific protocol other than the use of a liver specific AAV. It seems to me the protocol could be applied to any tissue or cell type where a specific targeting virus is available. There is no side-to-side comparison with other proteomic methods to demonstrate their method is superior to others. At the very least there needs to be an extensive comparison with literature to determine how this method compares with other methods for analyzing newly synthesized proteins.

Comments

1. It is stated in the Introduction without a citation, "BONCAT strategies were developed based on genetic manipulation of aminoacyl-tRNA synthetases in specific cell types of living animals followed by labeling of the nascent proteome with noncanonical amino acids." This is not correct. BONCAT was first introduced (2006) with the Met analog, azidohomoalanine (AHA), which uses the endogenous tRNA synthetase(1). It was reported in 2016 using azidonorleucine (ANL) with BONCAT that uses the genetic manipulation of Met tRNA synthetase(2).
2. There are at least two papers that need to be cited that analyzed newly synthesized proteins in the mouse liver(3, 4). They both employed AHA. Based on these papers, I do not know any reason why the newly synthesized liver proteome could not be specifically targeted with ANL using the Cre-lox genetic system either with breeding or AAV. Why was this approach not taken? How does your approach compare to ANL-BONCAT analysis of the liver?
3. The goal of the paper was stated to develop a comprehensive in vivo labeling strategy to decipher the nascent liver proteome in living with high efficiency and temporal resolution. The number of newly synthesized proteins identified was very low compared to the other SORT paper and the BONCAT literature. Also, the labeling was 7days or 20days which is longer than most BONCAT in vivo paper. Some labs are performing a single injection (24hrs) of label and identifying more newly synthesized proteins than reported here(5). Thus, I do not think this study has achieved high temporal resolution. Since there are more BONCAT papers than SORT, you should compare your results in terms of numbers and temporal resolution to the BONCAT literature. This needs to be discussed in the manuscript how your results compared to other newly synthesized proteome studies.
4. Please provide a reference for this statement about SORT: "direct use of this approach to label the nascent liver proteome has been unsuccessful, probably because of its low labeling efficiency in hepatocytes".
5. It is stated that "The engineered AlkKRS/4xPyITCUA pair significantly increased the incorporation efficiency compared to the original pair, as determined by flow cytometry (Fig. 1e and Fig. S1d)". Although "significantly" is used, there are no statistical tests shown. Please have a figure with errors bars and pvalues.
6. The study focuses on membrane proteins and devises a method to enrich them, but the modified protocol was never compared to the unmodified protocol. Thus, it is unknown if the modifications increase the identification of membrane

proteins. Furthermore, I did not understand the logic behind the modifications to improve membrane proteins identifications. It was proposed to focus on AA on the protein surface to make labeling and enrichment more efficient. If labeling is performed during translation, I do not understand how protein surface AA is more favorable. Also, the click reactions are performed with RIPA buffer to denature the proteins, so enrichment will not be limited to protein surface of AA. How does your coverage of membrane proteins compare to other published reports analyzing newly synthesized proteins?

7. Why was 7days labeling used for the first study and 20days used for the EtoH study? The increased labeling time did not appear to increase labeling efficiency. Why not?

8. It is stated: "The mice without AAV, AlkK or alcohol were used as blank control for endogenous biotinylated contaminant proteins." Since the lengthy EtOH treatment can cause large changes in the proteome, it would be better if the EtoH samples were used. Only a small fraction of the liver is used, so a blank control or enrichment can be performed on the same tissue without a click reaction. More importantly, what were the criteria for classifying a protein as a contaminant?

9. Figure 1h is not described correctly. PylT-KASM did not appear to be more efficient than other combinations as stated in the text. It seemed comparable to just using either A or M by itself.

10. The click reaction was optimized for the liver proteome, since initial attempts failed. You need to perform the click reaction in multiple tissues with your label to conclude it is a liver specific problem. For other studies, the click reaction was performed similarly for multiple tissues(3, 4). How do you explain this?

1. Dieterich DC, Link AJ, Graumann J, Tirrell DA, Schuman EM. Selective identification of newly synthesized proteins in mammalian cells using bioorthogonal noncanonical amino acid tagging (BONCAT). *Proc Natl Acad Sci U S A*. 2006;103(25):9482-7.

2. Mahdavi A, Hamblin GD, Jindal GA, Bagert JD, Dong C, Sweredoski MJ, et al. Engineered Aminoacyl-tRNA Synthetase for Cell-Selective Analysis of Mammalian Protein Synthesis. *J Am Chem Soc*. 2016;138(13):4278-81.

3. McClatchy DB, Ma Y, Liu C, Stein BD, Martinez-Bartolome S, Vasquez D, et al. Pulsed Azidohomoalanine Labeling in Mammals (PALM) Detects Changes in Liver-Specific LKB1 Knockout Mice. *J Proteome Res*. 2015;14(11):4815-22.

4. McClatchy DB, Martinez-Bartolome S, Gao Y, Lavalley-Adam M, Yates JR, 3rd. Quantitative analysis of global protein stability rates in tissues. *Sci Rep*. 2020;10(1):15983.

5. Schiapparelli LM, Xie Y, Sharma P, McClatchy DB, Ma Y, Yates JR, 3rd, et al. Activity-Induced Cortical Glutamatergic Neuron Nascent Proteins. *J Neurosci*. 2022;42(42):7900-20.

Version 1:

Reviewer comments:

Reviewer #1

(Remarks to the Author)

The authors have addressed a majority of my initial concerns. However, the authors still have the improper statistics for data with 3 groups (e.g., multiple groups) presented (Fig. 5f). If the data in Fig. 5f are normal a One-way ANOVA needs to be performed with a Tukey multiple comparison test. If the data are not normal then a Kruskal-Wallis test needs to be performed with a Dunn multiple comparison test. You can't present 3 groups and perform an unpaired student t-test between two of the groups. These statistical tests are available in the statistical software used by the authors. The multiple comparison test provides you the p-value for the head-to-head comparisons.

Reviewer #2

(Remarks to the Author)

I appreciate all the modifications to the manuscript to address my comments. Unfortunately, there are still problems with this manuscript that prevents it from being publishable.

1. Numerous comparisons were made with published BONCAT studies as requested, but modified text and figures are misleading, or I am puzzled by the results.

- It needs to be clearly stated in the introduction, the ANL-BONCAT can perform cell specific in vivo tissue analysis.

- Please state in the introduction what are the advantages or your reasons for choosing SORT over ANL-BONCAT. Are there any? Right now, it just states, "Only one approach based on aminoacyl-tRNA synthetase and noncanonical amino acids stood out", REF22. There has not been a direct comparison between BONCAT and SORT as far as I know. So, you can't state one is better than the other.

- In Fig.S1i, it shows your results compared to BONCAT HEK results in REF28. The BONCAT numbers in the graph do not match what is reported in this reference. In the reference, it is reported that 1932 ANL proteins and 1817 AHA proteins.

- In Fig.S3h, a comparison is presented between the nascent proteins identified with protocol from mouse liver and nascent proteins quantified in an AHA-BONCAT protein degradation study. The limit of quantification is higher than the limit of detection. In addition, the AHA-BONCAT study required proteins to be quantified in multiple animals and timepoints, which dramatically reduces the number of nascent proteins. Thus, this is an invalid comparison.

This study may be a more suitable comparison: PMID: 26445171

2. "Subsequent LC-MS/MS analysis of nascent proteomes, following streptavidin enrichment, further confirmed that this combination of residues not only increased total protein identification (Figure S1f) but also significantly enhanced membrane protein coverage (Fig.1i)."

The term significantly is used but there is no statistical test applied. More importantly, the percentage of membrane proteins identified in each condition is nearly identical in Fig.1i. So, this combination did increase total protein identification, but did not enhance the membrane protein coverage.

3. Figures were added to demonstrate examining protein turnover with the protocol. Protein turnover in the liver has been previously reported. Do your results correspond to previous results? Right now, there is nothing to support your claim that your protocol is "highly effective" to study protein turnover.

doi: 10.1016/j.cell.2025.02.021. Epub 2025 Mar 20.

doi: 10.1038/s41597-023-02537-w

4. An experiment was added where click reactions were performed with HEK cells in various tissue extracts(?) to test the compatibility of the tissue environment with the click reaction. It is difficult for me access this experiment, because there was no description of the experiment in the Methods.

5. There is no description in the methods on how membrane or transmembrane proteins were identified. A specific database?

Version 2:

Reviewer comments:

Reviewer #2

(Remarks to the Author)

The authors responded to comments and the manuscript is now acceptable for publication.

Our detailed responses to each question are listed below. For clarity, the original comments from each referee are shown in bold.

Reviewer #1 (Remarks to the Author):

The authors have developed a novel tool for the identification of tissue specific nascent proteome changes under different experimental conditions. This animal model is novel and very useful to the field. However, the authors have not done a full characterization of the dynamics of the nascent protein labeling in this study. Further, the incorporation of an alcohol model at the surface seems interesting but adds more questions since some important considerations were not made as detailed below. In summary I believe the animal model is novel and a great tool to investigate tissue specific nascent proteome but it needs more characterization. The way the alcohol model was performed I don't believe it is possible to interpret the results objectively because the no EtOH mice would have substantially more labeling as compared to the EtOH mice (e.g., the water only mice drink more than the 5% EtOH mice getting more AlkK).

We really appreciate the referee for reviewing our paper and providing insightful comments. As suggested, we have now included more characterization with additional experiments. Our point-by-point responses are detailed below, with corresponding changes highlighted in the revised manuscript.

Major Concerns:

1. It is not clear if identified labeled membrane proteins were selected for analysis or if only membrane proteins could be labeled. The description of the protein labeling mechanism for the 4 amino acids doesn't support the notion that only membrane proteins will be labeled.

We apologize for the lack of clarity in the original manuscript regarding this aspect. Our method is designed to label both the nascent cytosol and membrane proteome in cells and organs, while simultaneously achieving a notable enrichment of membrane proteins. The rationale for selecting the combination of K, A, S, and M as labeling amino acids is multifaceted: these residues exhibit (1) high distribution within the total proteins, (2) notable enrichment within transmembrane proteins, (3) high distribution in protein surface instead of interface. This combination therefore allows us to label the entire nascent proteome with a preferential enrichment of membrane proteins (red text on page 6).

To make it clearer, we have now included Figure 1f-g, which illustrates the distribution of each residue in both total and membrane proteins. Furthermore, Figure 1h demonstrates the combined labeling efficiency of K, A, S, and M in SORT-AC labeling. Furthermore, to unequivocally demonstrate the efficiency of membrane protein identification, we have performed comprehensive LC-MS/MS analysis of the labeled nascent proteins, the results of which are now presented in Figure 1i.

2. There is no methods section in the main body of the paper. This has to be included before a reviewer can assess or evaluate the manuscript.

We have now moved the detailed Methods section from the supplementary information to the main body of the revised manuscript (page 32-46). Thank you for pointing this out.

3. The dose of the AAV needs to be included.

Thank you for this important comment. The dose of AAV used in the experiments was 1×10^{12} viral particles per mouse in the legends of Figure 2 and 4, as well as within the dedicated method section of the revised manuscript.

4. There needs to be rationale for the EtOH model used. Currently the model uses 3 EtOH binges followed by 20 days of with background consumption of 5% EtOH. It is unclear what the rationale for the 3 binges are if the mice are euthanized 16 days later. Further, the dynamics of the nascent proteome would be of interest. How many of the nascent proteins are sustained for the 20 day period post 1st binge?

Thank you for your insightful comments. The rationale for employing the 3 EtOH binges followed by 20 days of background consumption of 5% EtOH is to effectively mimic human binge drinking patterns, where episodic high-intensity drinking is followed by a period of moderate alcohol consumption. This established procedure has been successfully utilized for constructing relevant alcoholic liver disease models (ref 1). Additionally, the 16-day post binge time with nascent proteome labeling was used to fully demonstrate the development of the alcoholic process in mouse liver, and to investigate the important nascent proteins during this process.

To comprehensively address the dynamics of nascent proteins, which are known to undergo rapid degradation, we performed detailed turnover investigations in both cellular and mouse liver models. In cellular experiments, SORT labeling followed by AlkK withdrawal revealed a rapid decrease in biotin signals from nascent proteins, which were almost undetectable at 30 hours (Figure 1j). Subsequent LC-MS/MS data revealed a significant reduction in the abundance of both total and membrane nascent proteins at 40 hours post-withdrawal (Figure 1k-m). To further verify the dynamics of nascent proteome in mice, we conducted a 7-day labeling period in mice, followed by cessation of AlkK administration and sequential euthanasia at different time points. Immunofluorescence results clearly demonstrated that nascent proteome signals in mouse liver rapidly diminished (Figure 2c-e).

We have included these data and discussion in the revised manuscript (red text on page 6-8).

Ref:

1. Wilkin, R. J., Lalor, P. F., Parker, R. & Newsome, P. N. Murine Models of Acute Alcoholic Hepatitis and Their Relevance to Human Disease. *Am J Pathol* 186, 748-760 (2016).

5. There needs to be a more thorough investigation to the impact of AlkK dose needed for consistent labeling. Did all the mice drink the same amount of water getting the same dose of AlkK? Mice will drink 5% EtOH but will drink less than water alone which would alter the AlkK dose between experimental groups. That is one reason the Lieber DeCarli diet and the method of pair feeding is used so that the amount of diet is equal between the pair-fed and alcohol-fed mice.

Thank you for the suggestion. To ensure uniform intake of AlkK across experimental groups, we implemented several rigorous measures. We utilized graduated drinking bottles, similar to those employed in the Lieber-DeCarli diet liquid feeding system, and conducted preliminary experiments to accurately estimate the average daily water intake of mice (approximately 5 mL/day) (Figure S5e). Throughout the experiments, the mice's daily water consumption was meticulously monitored.

6. An experiment should be performed to measure the rate of turnover of these nascent proteome in the liver. For example, provide the AlkK water for a week and euthanize a group of mice followed by replacing the water with no AlkK and euthanize mice daily until no nascent proteins are detectable. This is important characterization of the new animal model that is critical for future research.

Thank you for the valuable suggestion. As suggested, we have now performed turnover investigations in both cellular and mouse liver. Kindly refer to question 4.

In cellular experiments, we initiated SORT labeling and subsequently withdrew AlkK, collecting cells at various time points. We observed a rapid decrease in the biotin signals of nascent proteins (Figure 1j). To further verify the dynamics of nascent proteome in mouse tissue level, after 7-day labeling, we replaced the water with no AlkK and euthanize mice in different time points. The immunofluorescence result showed that the nascent proteome signals decreased rapidly (Figure 2c-e). The fast turnover observed in these results not only ensure the labeling specificity for nascent

proteome, but also provide valuable information for the application of this labeling strategy. These detailed data are now included in the revised manuscript (red text on page 6-8).

7. HepG2s are not a relevant cell line to study ethanol exposure. Primary mouse hepatocytes or AML-12 cells are more applicable in vitro models.

While HepG2 cells have been used in various ethanol-related studies as referenced below (Refs 1, 2), we acknowledge the reviewer's suggestion. We have now repeated the relevant ethanol treatment experiments using the more physiologically applicable AML12 cell line. As demonstrated in Figure S7f-h, Oil Red O and Nile Red staining in AML12 cells showed a significant increase in lipid accumulation after ethanol exposure, which was effectively reversed by treatment with Triacsin C. These results are highly consistent with our initial findings in HepG2 cells (red text on page 14).

Ref:

1. Petagine, L., Zariwala, M.G., Somavarapu, S. et al. Oxidative stress in a cellular model of alcohol-related liver disease: protection using curcumin nanoformulations. *Sci Rep* 15, 7752 (2025).
2. Correnti, J.M., Gottshall, L., Lin, A. et al. Ethanol and C2 ceramide activate fatty acid oxidation in human hepatoma cells. *Sci Rep* 8, 12923 (2018).

8. There should be a more thorough phenotyping of the experimental mice other than liver enzymes and steatosis.

We have now expanded the phenotyping of our experimental mice. The revised manuscript includes data on body weight changes (Figure S5c), comprehensive H&E staining of liver tissues (Figure S5b), and qPCR measurements of key inflammation-related gene expression (Figure S5d). These additions provide a more thorough characterization of the physiological impact of our experimental conditions (red text on page 11).

Minor concerns:

1. Title needs to be altered to remove “alcoholic”. For example, Nascent liver proteome reveals novel enzymes and transcription regulators under physiological and conditions of alcohol exposure.

We have revised our title to: "Nascent liver proteome reveals novel enzymes and transcription regulators under physiological and alcohol exposure conditions" to more accurately and broadly convey the focus of our research. Thank you!

2. The raw proteome data is not found with the provided ID at the repository noted by the authors and the hyperlink requires a password. If the data is under embargo it needs to be lifted and verified.

We have now made our proteome data publicly accessible. The data can be directly accessed using the provided identifier in the Data Availability section of the revised manuscript, without the need for a password (page 46).

3. Since EtOH is known to affect protein translation it may be important to note nascent proteins that were enriched in the No EtOH group vs EtOH group.

We have now incorporated a comparison of the dynamic changes in nascent proteome between the no-EtOH and EtOH groups in the revised manuscript (Figure 4f). Furthermore, we have presented the pathway enrichment analysis for EtOH-enriched proteins in Figure 4g. This data clearly suggests that protein translation efficiency is significantly elevated during ethanol treatment.

4. Individual data points are not shown for sample size $n = 10$.

Thank you for the suggestion. To improve data visualization and transparency, we have now labeled each individual data point in all relevant figures where the sample size is $n \leq 10$ (e.g. Figure 4c, Figure 5b, Figure 5f and so on).

5. No statistic test are provided for figures (fig. 5f) and where they are presented they are not appropriate.

Thank you for pointing this out. We have now thoroughly reviewed and included appropriate statistical analysis for Figure 5f and all other relevant figures.

Reviewer #2 (Remarks to the Author):

This manuscript presents a proteomic method to identify newly synthesized proteins from the liver with a focus on membrane proteins. Their approach is the modification of a previous protocol termed SORT. Interesting novel biological findings were discovered when they applied their method to an EtOH induced injury mouse model. It was unclear, however, why this was a liver specific protocol other than the use of a liver specific AAV. It seems to me the protocol could be applied to any tissue or cell type where a specific targeting virus is available. There is no side-to-side comparison with other proteomic methods to demonstrate their method is superior to others. At the very least there needs to be an extensive comparison with literature to determine how this method compares with other methods for analyzing newly synthesized proteins.

We really appreciate the referee for reviewing our paper and providing insightful comments. We have addressed all the points with additional experiments and discussion.

Comments

1. It is stated in the Introduction without a citation, “BONCAT strategies were developed based on genetic manipulation of aminoacyl-tRNA synthetases in specific cell types of living animals followed by labeling of the nascent proteome with noncanonical amino acids.” This is not correct. BONCAT was first introduced (2006) with the Met analog, azidohomoalanine (AHA), which uses the endogenous tRNA synthetase (1). It was reported in 2016 using azidonorleucine (ANL) with BONCAT that uses the genetic manipulation of Met tRNA synthetase(2).

We appreciate you highlighting this historical inaccuracy. We have now revised the statement in the Introduction to accurately reflect the development of BONCAT strategies, clarifying that early BONCAT approaches utilized endogenous tRNA synthetases with methionine analogs like azidohomoalanine (AHA), and later advancements incorporated genetic manipulation of aminoacyl-tRNA synthetases for site-specific or expanded noncanonical amino acid incorporation (red text on page 3). Relevant literature references have been added to ensure accuracy.

2. There are at least two papers that need to be cited that analyzed newly synthesized proteins in the mouse liver (3, 4). They both employed AHA. Based on these papers, I do not know any reason why the newly synthesized liver proteome could not be specifically targeted with ANL using the Cre-lox genetic system either with breeding or AAV. Why was this approach not taken? How does your approach compare to ANL-BONCAT analysis of the liver?

Thank you for this insightful question. While ANL-BONCAT combined with Cre-LoxP offers a promising approach for tissue-specific labeling, we found that its practical application for comprehensive liver proteome analysis, especially for membrane proteins, has limitations due to its reliance on methionine incorporation. Methionine's relatively low abundance in mammalian proteins and its frequent N-terminal cleavage can lead to suboptimal labeling efficiency (Figure 1h-i). Importantly, methionine content is particularly low in transmembrane proteins (Figure 1g). In contrast, our optimized SORT-based approach, which leverages broader amino acid incorporation, offers greater flexibility and significantly higher labeling efficiency across diverse protein classes, particularly for membrane proteins. As elaborated in our response to Comment 3, our method demonstrates superior identification rates for both total and membrane proteins compared to reported BONCAT datasets (Figure S3h).

3. The goal of the paper was stated to develop a comprehensive in vivo labeling strategy to decipher the nascent liver proteome in living with high efficiency and temporal resolution. The number of newly synthesized proteins identified was very low compared to the other SORT paper and the BONCAT literature. Also, the labeling was 7days or 20days which is longer than most BONCAT in vivo paper. Some labs are performing a single injection (24hrs) of label and identifying more newly synthesized proteins than reported here(5). Thus, I do not think this

study has achieved high temporal resolution. Since there are more BONCAT papers than SORT, you should compare your results in terms of numbers and temporal resolution to the BONCAT literature. This needs to be discussed in the manuscript how your results compared to other newly synthesized proteome studies.

We appreciate this crucial feedback regarding the scope and resolution of our method. We agree that a robust comparison with existing literature is essential to contextualize our findings and highlight the unique strengths of our SORT-AC strategy. We have now performed additional analyses (both in cellular and tissue levels) and revised the manuscript to address these points.

We first compared our strategy's performance with BONCAT (using ANL and AHA) in cellular systems (ref 1). Interestingly, the total number of nascent proteins identified was largely comparable across all three methods. However, our SORT-AC strategy consistently identified significantly more membrane proteins (Figure S1i), underscoring its inherent advantage for investigations targeting this specific protein class. This highlights a key strength of our approach for membrane protein research.

At the tissue level, we acknowledge that the 24-hour short-term labeling studies often rely on invasive intraperitoneal ANL injections, which, while effective for acute labeling, may not be suitable for the chronic models we investigate. While we were unable to access certain BONCAT liver datasets (ref 2), we successfully compared our SORT-AC data with a publicly available 4-day AHA-labeling dataset from liver (ref 3), which shares a similar workflow to our study. As shown in the newly added Figure S3h, our method detected a significantly greater number of total nascent proteins and, critically, a higher proportion of membrane proteins compared to the BONCAT dataset.

While our 7-day and 20-day labeling periods may appear longer than some acute BONCAT studies, they were deliberately chosen to investigate chronic physiological and pathological changes, such as those induced by sustained ethanol consumption. To further address the aspect of "temporal resolution" and the steady-state labeling observed, we have performed a dynamic analysis in the culture cells and mouse liver (Figure 2c-e). We have now included the above information in the revised manuscript (red text on page 6-9).

Ref:

1. Schiapparelli, L. M. et al. Activity-Induced Cortical Glutamatergic Neuron Nascent Proteins. *J Neurosci* 42, 7900-7920 (2022).
2. McClatchy, D. B. et al. Pulsed Azidohomoalanine Labeling in Mammals (PALM) Detects Changes in Liver-Specific LKB1 Knockout Mice. *J Proteome Res* 14, 4815-4822 (2015).
3. McClatchy, D. B., Martínez-Bartolomé, S., Gao, Y., Lavallée-Adam, M. & Yates, J. R., 3rd. Quantitative analysis of global protein stability rates in tissues. *Sci Rep* 10, 15983 (2020).

4. Please provide a reference for this statement about SORT: “direct use of this approach to label the nascent liver proteome has been unsuccessful, probably because of its low labeling efficiency in hepatocytes”.

We have now revised this statement and provided experimental evidence to support our claim that the initial SORT procedure exhibited low labeling efficiency in hepatocytes and that the liver environment presented challenges for direct application without optimization. As shown in the newly added Supplementary Figure S3a, initial, unoptimized SORT procedures indeed failed to efficiently capture labeled proteins. Furthermore, control experiments involving mixing labeled cells with extracts from various mouse organs (Figure S3b-c) confirmed that the liver environment, due to its complex metabolic profile, significantly interfered with the click reaction efficiency.

5. It is stated that “The engineered AlkKRS/4xPyITCUA pair significantly increased the incorporation efficiency compared to the original pair, as determined by flow cytometry (Fig. 1e and Fig. S1d)”. Although “significantly” is used, there are no statistical tests shown. Please have a figure with errors bars and pvalues.

Thank you for pointing this out. As suggested, we have repeated the experiment three times and applied statistical tests for error bars and p-values (Figure S1d).

6. The study focuses on membrane proteins and devises a method to enrich them, but the modified protocol was never compared to the unmodified protocol. Thus, it is unknown if the modifications increase the identification of membrane proteins. Furthermore, I did not understand the logic behind the modifications to improve membrane proteins identifications. It was proposed to focus on AA on the protein surface to make labeling and enrichment more efficient. If labeling is performed during translation, I do not understand how protein surface AA is more favorable. Also, the click reactions are performed with RIPA buffer to denature the proteins, so enrichment will not be limited to protein surface of AA. How does your coverage of membrane proteins compare to other published reports analyzing newly synthesized proteins?

We apologize for the lack of clarity in the original manuscript regarding this aspect. Our method is designed to label both the nascent cytosol and membrane proteome in cells and organs, while simultaneously achieving a notable enrichment of membrane proteins. The rationale for selecting the combination of K, A, S, and M as labeling amino acids is multifaceted: these residues exhibit (1) high distribution within the total proteins, (2) notable enrichment within transmembrane proteins, (3) high distribution in protein surface instead of interface (Figure 1f-g). This combination therefore allows us to label the entire nascent proteome with a preferential enrichment of membrane proteins. We have now modified the description in the manuscript to avoid any ambiguity (red text on page 6).

Our additional experiments, including streptavidin-blot (Figure S1f) and LC-MS/MS analysis (Figure 1i), confirm that the KASM residue combination significantly improves both total protein identification and, crucially, membrane protein coverage. As discussed in response to Question #3, our method demonstrates superior coverage of membrane proteins compared to other published reports (Figure S1i and Figure S3h). We have revised the manuscript to provide clearer explanations (red text on page 7 and 9).

7. Why was 7days labeling used for the first study and 20days used for the EtoH study? The increased labeling time did not appear to increase labeling efficiency. Why not?

The 20-day labeling period for the alcohol-induced liver injury model was necessary to better mimic the prolonged alcohol consumption associated with this condition. While labeling intensity plateaued after 7 days, indicating a rapid and effective labeling of the nascent proteome, this steady state reflects the dynamic balance between protein synthesis and degradation. Our new dynamic analysis in mouse liver (Figure 2c-e) further demonstrates that nascent protein signals significantly decrease within 7 days after label withdrawal, supporting the rapid turnover and confirming that our 7-day labeling is sufficient for comprehensive nascent proteome profiling.

8. It is stated: “The mice without AAV, AlkK or alcohol were used as blank control for endogenous biotinylated contaminant proteins.” Since the lengthy EtOH treatment can cause large changes in the proteome, it would be better if the EtoH samples were used. Only a small fraction of the liver is used, so a blank control or enrichment can be performed on the same tissue without a click reaction. More importantly, what were the criteria for classifying a protein as a contaminant?

Thank you for your valuable advice regarding controls. We employed two distinct control groups:

(1) Mice without AAV, AlkK, or alcohol treatment were used as a blank control to identify endogenous biotinylated contaminant proteins.

(2) Mice undergoing the same treatment as test groups but without EtOH consumption served as a physiological control, reflecting the nascent proteome under steady-state conditions. Our primary focus was on the biologically significant comparison between non-ethanol and ethanol-treated groups. We purposely separated liver fractions from different mice for control and test to minimize

individual mouse effects.

While endogenous biotinylated contaminants were present in both no-EtOH and EtOH groups, our data analysis workflow involved rigorous filtering of enriched targets against a contaminant database (CRAPome) and comparison with the blank control group to prevent false positives.

9. Figure 1h is not described correctly. PyIT-KASM did not appear to be more efficient than other combinations as stated in the text. It seemed comparable to just using either A or M by itself.

We sincerely apologize for this misrepresentation and the resulting confusion in Figure 1h. We have now repeated this experiment three times and performed grayscale intensity quantification. The results demonstrated that PyIT-KASM exhibits superior labeling efficiency compared to the use of individual amino acids (A or M alone) (Figure 1h).

Meanwhile, to further clarify the labeling ability among different residue combinations, we have now performed enrichment (Figure S1f) and identification by LC-MS/MS (Figure 1i). The result showed that the combination of KASM actually increases the labeling performance of both total proteins and membrane proteins.

10. The click reaction was optimized for the liver proteome, since initial attempts failed. You need to perform the click reaction in multiple tissues with your label to conclude it is a liver specific problem. For other studies, the click reaction was performed similarly for multiple tissues (3, 4). How do you explain this?

Thank you for your suggestion. We have performed a new set of experiments where we applied our unoptimized click reaction conditions to HEK293T cells that had been labeled with SORT, which were then mixed with extracts from various mouse organs (heart, kidney, lung, small intestine, and liver) (Figure S3b-c). The results showed a marked reduction in reaction signals in the liver and small intestine groups (Figure S3b-c). This experiment provides compelling evidence that the complexity of the liver's biochemical milieu indeed poses a significant hurdle for efficient click reaction, thus justifying the extensive optimization steps we undertook for this tissue in the current study (red text on page 8-9).

Our detailed responses to each question are listed below. For clarity, the original comments from each referee are shown in bold.

Reviewer #1 (Remarks to the Author):

The authors have addressed a majority of my initial concerns. However, the authors still have the improper statistics for data with 3 groups (e.g., multiple groups) presented (Fig. 5f). If the data in Fig. 5f are normal a One-way ANOVA needs to be performed with a Tukey multiple comparison test. If the data are not normal then a Kruskal-Wallis test needs to be performed with a Dunn multiple comparison test. You can't present 3 groups and perform an unpaired student t-test between two of the groups. These statistical tests are available in the statistical software used by the authors. The multiple comparison test provides you the p-value for the head-to-head comparisons.

Thank you for this crucial comment. We have thoroughly re-evaluated all datasets and performed statistical analyses tailored to their distributions. For datasets exhibiting a normal distribution, a one-way ANOVA was conducted, followed by Tukey's multiple comparison test. For non-normally distributed datasets, a Kruskal-Wallis test with Dunn's multiple comparison test was used. All relative figures and legends have been updated to reflect these analyses, and exact P values, adjusted for multiple comparisons, are now presented based on these multiple comparisons (Fig 5f, FigS1d, FigS5g-h, FigS7g, i-k).

Reviewer #2 (Remarks to the Author):

I appreciate all the modifications to the manuscript to address my comments.

Unfortunately, there are still problems with this manuscript that prevents it from being publishable.

1. Numerous comparisons were made with published BONCAT studies as requested, but modified text and figures are misleading, or I am puzzled by the results.

• It needs to be clearly stated in the introduction, the ANL-BONCAT can perform cell specific in vivo tissue analysis.

Thank you once again for your meticulous review and invaluable feedback on our manuscript. We really appreciate your continued engagement and have carefully made revision based on your comments. We have now revised the manuscript to ensure accuracy and to provide clearer explanations.

• Please state in the introduction what are the advantages or your reasons for choosing SORT over ANL-BONCAT. Are there any? Right now, it just states, “Only one approach based on aminoacyl-tRNA synthetase and noncanonical amino acids stood out”, REF22. There has not been a direct comparison between BONCAT and SORT as far as I know. So, you can’t state one is better than the other.

You are correct that a direct head-to-head comparison between BONCAT and SORT, as far as we know, has not been performed, and therefore, stating one is "better" is inappropriate. Both BONCAT and SORT are uniquely capable of identifying the nascent proteome from spatially and genetically defined regions of living animals. We selected SORT for further optimization in our study, primarily because its substrate, AlkK, is readily synthesized in our laboratory. We have revised the relevant section of the manuscript to clarify the rationale (red text on page 4-5).

• In Fig.S1i, it shows your results compared to BONCAT HEK results in REF28. The BONCAT numbers in the graph do not match what is reported in this reference. In the reference, it is reported that 1932 ANL proteins and 1817 AHA proteins.

We sincerely appreciate you pointing this out. We have revisited the reference (REF28, PMID: 36261270) to verify the numbers. You are correct regarding the numbers reported in the main text of PMID: 36261270 (1,932 ANL proteins and 1,817 AHA proteins). However, upon re-examining the supplementary data file provided by the authors of Reference 28 (<https://www.jneurosci.org/content/jneuro/42/42/7900/DC1/embed/inline-supplementary-material-1.xlsx?download=true>), we found that the number of proteins listed in the dataset was 1,806 for ANL and 1,708 for AHA. To ensure consistency and for our specific analysis of membrane proteins, we utilized the protein numbers directly from this supplementary dataset. The original dataset “inline-supplementary-material-1.xlsx” is attached for verification (kindly refer to Reviewer only file 1). Nevertheless, we have removed all head-to-head comparison from the revised manuscript. Kindly refer to our response to the next comment.

• In Fig.S3h, a comparison is presented between the nascent proteins identified with protocol from mouse liver and nascent proteins quantified in an AHA-BONCAT protein degradation study. The limit of quantification is higher than the limit of detection. In addition, the AHA-BONCAT study required proteins to be quantified in multiple animals and timepoints, which

dramatically reduces the number of nascent proteins. Thus, this is an invalid comparison.

This study may be a more suitable comparison: PMID: 26445171

We appreciate your insight regarding the comparisons. We agree that direct head-to-head comparisons between BONCAT and SORT are not appropriate for this manuscript, especially since we didn't perform such a comparison within our study. Both SORT and BONCAT are powerful techniques for identifying nascent proteomes from specific regions in living animals. To ensure clarity and avoid any misleading interpretations, we've removed all comparisons from the revised manuscript, including the data previously presented in Figure S1i.

We attempted to compare our results with PMID: 26445171, as you suggested. Unfortunately, the data repository link provided in that article (<http://sealion.scripps.edu/pint/?project=f302f024b4d5aa1f>) is currently inaccessible.

To address your broader point about differences in protein identification across various studies, we've compiled a detailed "Reviewer Only Table." This table, provided below, outlines the key experimental parameters, such as sample processing, mass spectrometry analysis, and LC setup, from our study, PMID: 26445171, and the cell-based comparison papers (PMID: 36261270, PMID: 25117199). This highlights the substantial methodological differences that exist. We believe these variations in sample preparation and data acquisition likely introduce discrepancies far greater than any intrinsic differences between the BONCAT and SORT methods themselves, making direct numerical comparisons inappropriate. Therefore, we have removed all head-to-head comparison from the revised manuscript.

[editorial note: table redacted]

Reviewer only table. The comparison of methods details among this study and suggested references. (kindly refer to Reviewer only file 2)

2. “Subsequent LC-MS/MS analysis of nascent proteomes, following streptavidin enrichment, further confirmed that this combination of residues not only increased total protein identification (Figure S1f) but also significantly enhanced membrane protein coverage (Fig.1i).”

The term significantly is used but there is no statistical test applied. More importantly, the percentage of membrane proteins identified in each condition is nearly identical in Fig.1i. So, this combination did increase total protein identification, but did not enhance the membrane protein coverage.

We agree and have revised the corresponding sections in the manuscript to provide a clearer description (red text on page 6). Thank you!

3. Figures were added to demonstrate examining protein turnover with the protocol. Protein turnover in the liver has been previously reported. Do your results correspond to previous results? Right now, there is nothing to support your claim that your protocol is “highly effective” to study protein turnover.

doi: 10.1016/j.cell.2025.02.021. Epub 2025 Mar 20.

doi: 10.1038/s41597-023-02537-w

Thank you for your insightful comments. We employed click chemistry–based immunofluorescence labeling for a semi-quantitative analysis of nascent protein signals. Our data indicate a protein turnover half-life of approximately 3–5 days in mouse liver, which aligns well with the reported value of 3.28 days in existing literature (Ref). We have revised the corresponding statements in the manuscript to include this comparison and have cited the relevant reference (red text on page 8).

Ref:

1.Li W, et al. Turnover atlas of proteome and phosphoproteome across mouse tissues and brain regions. Cell 188, 2267-2287.e2221 (2025).

4. An experiment was added where click reactions were performed with HEK cells in various tissue extracts(?) to test the compatibility of the tissue environment with the click reaction. It is difficult for me access this experiment, because there was no description of the experiment in the Methods.

Thank you for pointing this out. The detailed experimental procedures have now been included in the revised Methods section (kindly refer to Method section under “8. Analysis of SORT-labeled proteome in HEK293T with click chemistry”) (red text on page 39-40).

5. There is no description in the methods on how membrane or transmembrane proteins were identified. A specific database?

Thank you for your suggestion. We identified membrane proteins and transmembrane proteins using the UniProtKB/Swiss-Prot database. We have updated the Methods section to clarify this (red text on page 12 and page 45).